Corrected: Author correction

# Zebrafish and medaka offer insights into the neurobehavioral correlates of vertebrate magnetoreception

Ahne Myklatun[1,2,3], Antonella Lauri [1,2,3], Stephan H.K. Eder[4], Michele Cappetta[1,2,3], Denis Shcherbakov[5], Wolfgang Wurst[2], Michael Winklhofer [6,7] & Gil G. Westmeyer[1,2,3]

An impediment to a mechanistic understanding of how some species sense the geomagnetic field ("magnetoreception") is the lack of vertebrate genetic models that exhibit well-characterized magnetoreceptive behavior and are amenable to whole-brain analysis. We investigated the genetic model organisms zebrafish and medaka, whose young stages are transparent and optically accessible. In an unfamiliar environment, adult fish orient according to the directional change of a magnetic field even in darkness. To enable experiments also in juveniles, we applied slowly oscillating magnetic fields, aimed at generating conflicting sensory inputs during exploratory behavior. Medaka (but not zebrafish) increase their locomotor activity in this assay. Complementary brain  activity mapping reveals neuronal activation in the lateral hindbrain during magnetic stimulation. These comparative data support magnetoreception in teleosts, provide evidence for a light-independent mechanism, and demonstrate the usefulness of zebrafish and medaka as genetic vertebrate models for studying the biophysical and neuronal mechanisms underlying magnetoreception.

[1] Institute of Biological and Medical Imaging, Helmholtz Zentrum München, Ingolstädter Landstrasse 1, 85764 Neuherberg, Germany. [2] Institute of Developmental Genetics, Helmholtz Zentrum München, Ingolstädter Landstrasse 1, 85764 Neuherberg, Germany. [3] Department of Nuclear Medicine, Technical University of Munich, Ismaninger Strasse 22, 81675 Munich, Germany. [4] Department of Earth- and Environmental Sciences Section Geophysics, Ludwig Maximilian University of Munich, Theresienstrasse 41, 80333 Munich, Germany. [5] Institute of Zoology 220, University of Hohenheim, 70593 Stuttgart, Germany. [6] Institute for Biology and Environmental Sciences IBU, Carl von Ossietzky University of Oldenburg, Carl-von-Ossietzky-Strasse 9-11, 26129 Oldenburg, Germany. [7] Research Center Neurosensory Science, Carl von Ossietzky Universität Oldenburg, Oldenburg, D-26111, Germany. Ahne Myklatun and Antonella Lauri contributed equally to this work.  Correspondence and requests for materials should be addressed to G.G.W. (email: gil.westmeyer@tum.de)

The geomagnetic field (GMF) varies systematically across the surface of the Earth in polarity (direction, North and South), inclination (angle between field lines and the Earth's surface), and intensity, offering a spatial and directional reference frame for orientation and navigation. Several animals were reported to sense the Earth's magnetic field ("magnetoreception", Fig. 1)[1]. Spiny lobsters may use the sense to return home[2,3], migratory birds to find their destination[4], mole rats show a preferred geomagnetic orientation when they build their nest[5,6], and cockroaches become more active in slowly oscillating magnetic fields[7,8].

Despite the widespread occurrence of magnetoreception across different phyla (Fig. 1), the biophysical and neuronal mechanisms underlying magnetoreception are poorly understood. Two main hypotheses on the mechanistic basis of the magnetic sense exist. (i) Magnetic fields can bias photochemical reactions involving radical pairs ("radical pair hypothesis")[9]. This might be physically realized in cryptochromes of the retina, and translated into neuronal signals under short-wavelength light (400–500 nm)[10–12].

Behavioral evidence exists for different species supporting such a light-dependent sense[13–16] and disruption of magnetoreceptive behavior via weak magnetic fields in the radiofrequency range seems to further support the radical pair mechanism[17,18] (Fig. 1a). In addition, genetic manipulations in insects indicate the involvement of cryptochromes in magnetoreception[7,19,20]. (ii) Alternatively, ferrimagnetic material (e.g. biomineralized magnetite) interacting with the GMF could exert forces on mechanosensitive cellular structures, which can then be transduced into neuronal signals ("magnetite hypothesis")[21–23] (Fig. 1a). This putative mechanism can work independently of light and is consistent with behavioral data from several animals[2,3,24–27]. Furthermore, altered orientation behavior after treatment with a strong magnetic pulse indicates the involvement of magnetic material in birds[28,29]. It is interesting to note that in some species the two mechanisms seem to coexist and can detect different parameters of the GMF[28,30–36] (Fig. 1a).

Candidate brain circuits of magnetoreception, such as the trigeminal brainstem complex in rainbow trouts[37,38], European

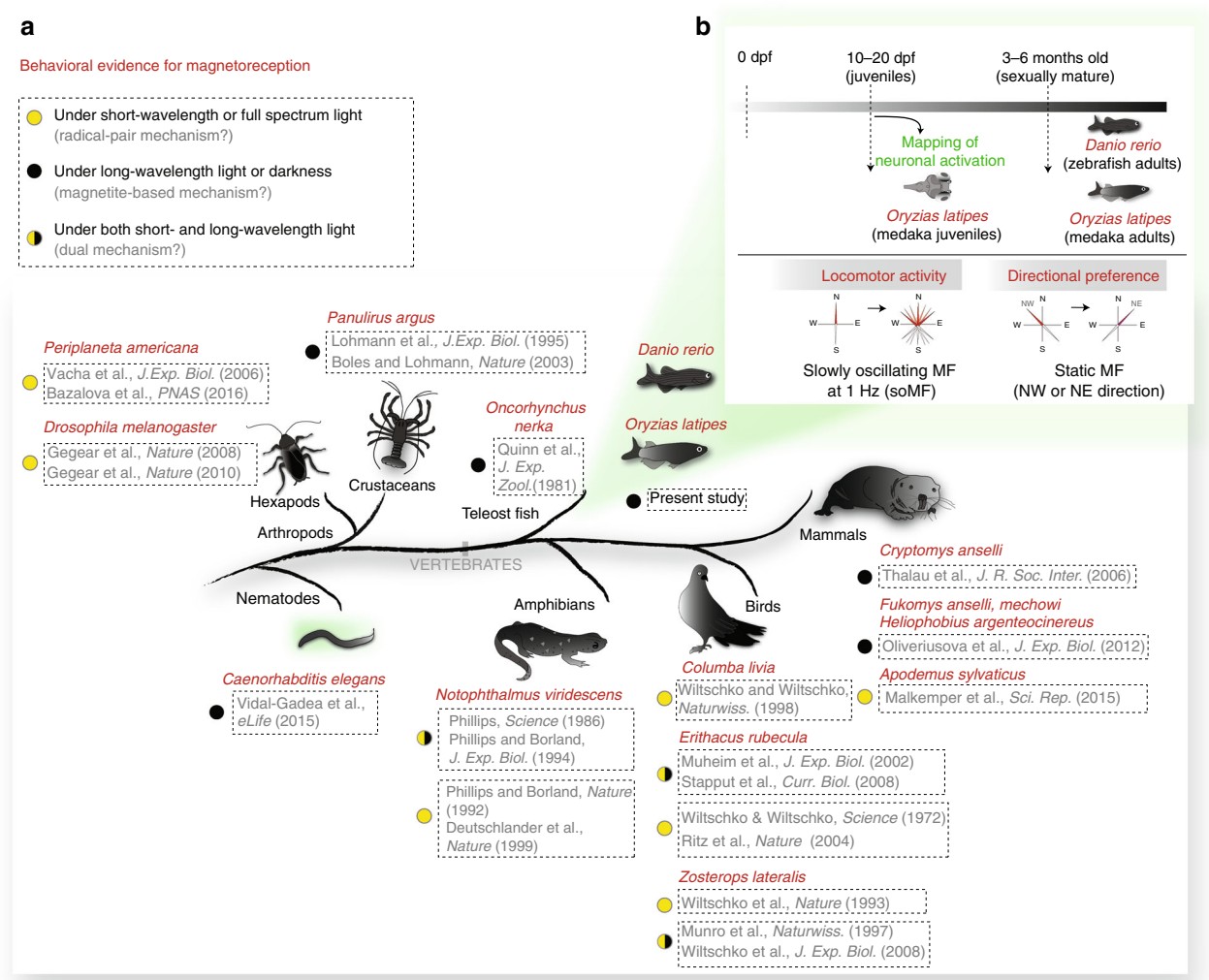

**Fig. 1** Behavioral evidence for magnetoreception across the animal kingdom. **a** Schematic summarizing the major experimental evidence for magnetoreception in invertebrates and vertebrates as reported in the selected studies. The colored circles next to the references indicate whether the behavioral experiments provide evidence for a light-dependent mechanism (i.e. consistent with the "radical pair hypothesis", yellow), for a light-independent mechanism (i.e. working under long-wavelength light or darkness, not consistent with a radical pair mechanism but consistent with the "magnetite hypothesis", black) or for the presence of a dual mechanism (yellow/black). The green shadow indicates genetic model organisms that are accessible by whole-brain optical imaging. **b** Design of the present study performed on zebrafish and medaka at different developmental stages. Both juveniles and sexually mature fish were studied by customized behavioral assays ("locomotor activity" for juveniles and "directional preference" for sexually mature fish). Neuronal activation during the "locomotor activity" assay was mapped in medaka juveniles. dpf: days post fertilization

robins[39] and bobolinks[40], the visual system in mole rats[41], and the ear and vestibular system in pigeons[42,43], have been proposed. However, the tedious brain sectioning or invasive recordings necessary in these non-genetic and non-transparent vertebrate models pose serious challenges for unraveling the precise circuit underlying magnetoreception.

Due to their small size, relative transparency, as well as their amenability to genetic manipulations, the teleosts zebrafish (*Danio rerio*) and medaka (*Oryzias latipes*) can be ideal models for studying the biophysical and neuronal basis of vertebrate magnetoreception. Comparable features are found only in the nematode *Caenorhabditis elegans*, which however may employ different mechanisms for magnetoreception as compared to vertebrates[44].

Recent reports have suggested that zebrafish are magnetoreceptive[45–48], but no knowledge regarding the mechanism or the neuronal circuits is yet available. In this work, we sought to conduct a comparative study in two teleost models, to test (i) whether light-independent magnetoreception is present, and (ii) whether a behavioral assay could be established in young fish that is compatible with neuronal activity mapping across the whole-brain (Fig. 1b). We chose medaka for this comparison because, similar to zebrafish, it is transparent at early stages and easy to culture[49], but has a less redundant genome[50] which is ideal for genetic studies. Furthermore, in contrast to zebrafish, medaka seems to migrate between fresh and seawater, a behavior for which navigational capabilities are beneficial.

Our experiments show that the direction of the magnetic field influences the orientation of zebrafish and medaka adults even in the absence of visible light, suggesting the use of a light-independent sensing mechanism. Furthermore, we established a behavioral assay for juvenile fish, which indicates that magneto-sensitivity is already present in young medaka, but not in young zebrafish. Brain activity mapping furthermore showed neuronal activation of the posterior lateral hindbrain in young medaka in response to stimulation with oscillating magnetic fields. This comparative study suggests that magnetoreception might be a common feature of teleosts and that zebrafish and medaka are attractive vertebrate models for future research on the biophysical mechanism and neuronal substrates of magnetoreception.

## Results

**Zebrafish and medaka orient in response to the MF direction.** To assess magnetoreception in sexually mature fish, we employed a behavioral assay similar to Takebe et al.[46], who showed directional preference of genetic cohorts of zebrafish in a magnetic field (MF) after release from the center of a circular arena (Fig. 2a). We chose an intra-subjective design in which each fish was tested twice, in two different magnetic conditions, obtained by setting the horizontal component of GMF 45° either towards East (NE) or West (NW), using two pairs of Helmholtz coils (Fig. 2a, b and Supplementary Fig. 1). This experimental design ensured that the current running through the coils was the same (but of opposite direction in the E−W coil pair) in both conditions. In distinction to previous studies in zebrafish[46–48], the behavioral response could thus be confidently assigned to the change of the direction of the MF. In addition, we tested fish in presence or absence of short-wavelength light (Fig. 2c). For zebrafish, we then analyzed whether the change in bearing of each fish followed the 90° field deflection that we applied between conditions. Following Takebe et al.[46], bearing (BE) was defined as the line between the center of the arena and the point where the fish first crossed a virtual circle (Fig. 2d, Supplementary Fig. 2). We found that under illumination with white light (WL, Fig. 2c) zebrafish significantly changed their bearing such that the

distribution of angular differences between the two magnetic conditions (NE–NW) showed a mean axis which was consistent with the 90° deflection of the MF (Fig. 2e).

Next, we tested the fish under infrared illumination (IR, 1060 nm; Fig. 2c), which is not visible for many fish species, including zebrafish[51]. Besides avoiding potentially confounding visual cues, this experimental condition further aimed at assessing the presence of a light-independent mechanism of magnetoreception, postulated by the "magnetite hypothesis", discussed above. To avoid possible behavioral effects due to a change in illumination between acclimation and testing, one group of zebrafish was additionally acclimated to darkness for 60 min (D-IR group). This pre-adaptation period should further exclude any persisting photo-induced MF sensing, given the sub-second lifetime of the radical pairs in cryptochromes, which is assumed to determine the signaling states updating neuronal processing during magnetoreception[11,12]. Moreover, behavioral experiments in birds tested under long wavelengths have suggested that exposure to darkness for an hour prior to testing is sufficient to prevent light-dependent magnetoreception[52].

When we assessed the directional preference in this group of zebrafish, we found a significant change of the bearing, consistent with the 90° deflection of the MF (Fig. 2f). Interestingly, the distribution of the individual angular differences in the D-IR group seemed to be polar, and was significantly different from that observed in light (WL vs. D-IR, Watson $U^2$: $p < 0.05$), indicating that two different mechanisms might be at play. Furthermore, when zebrafish were tested twice in the D-IR condition but without changing the direction of the MF, they showed a mean reorientation angle whose axis was consistent with a 0° deflection of the field (Supplementary Fig. 3). Without acclimation of the fish to darkness prior to testing under IR illumination (IR group), we observed a large scatter in the data resulting in a distribution that was not significantly clustered (Supplementary Fig. 4). This may have been due to the abrupt change in illumination between acclimation (in WL) and testing (under IR), which likely constituted a stress factor for the fish[53,54].

When testing medaka under IR, we first noted that they exhibited a different swimming behavior. While zebrafish followed a rather linear trajectory towards the perimeter of the arena to then swim along the wall (thigmotaxis[55], Supplementary Fig. 5, Supplementary Table 1, Supplementary Movie 1), medaka explored the arena by swimming in tight circles ("looping behavior") (Supplementary Fig. 6, Supplementary Table 1, Supplementary Movie 2), as also shown in rodent exploratory behavior[56]. To quantify whether the observed swimming behavior was influenced by the deflection of the MF (change in declination), we analyzed the spatial preference of medaka (SP) by determining the segment of the arena in which the fish spent the majority of time (Fig. 2g, Supplementary Fig. 7). We found that medaka adapted to darkness for 60 min and tested in IR (D-IR group, Fig. 2c) showed a significant axial change in spatial preference, compatible with the 90° deflection of the MF (Fig. 2h). Without acclimation to darkness prior to testing, we again observed a large variation in the directional preference change of an independent cohort of medaka (IR group, Supplementary Fig. 8).

Previous studies have shown that a directional preference relative to the MF exists within groups of zebrafish tested only once[46,47]. When we analyzed the initial trial for each fish in our experiments to assess the group response in a comparable fashion, we observed that zebrafish of the D-IR group (AB strain) showed a significant axial distribution in their directional preference (Supplementary Fig. 9a). This is similar to the result reported by Takebe et al. for the EKK zebrafish strain. The

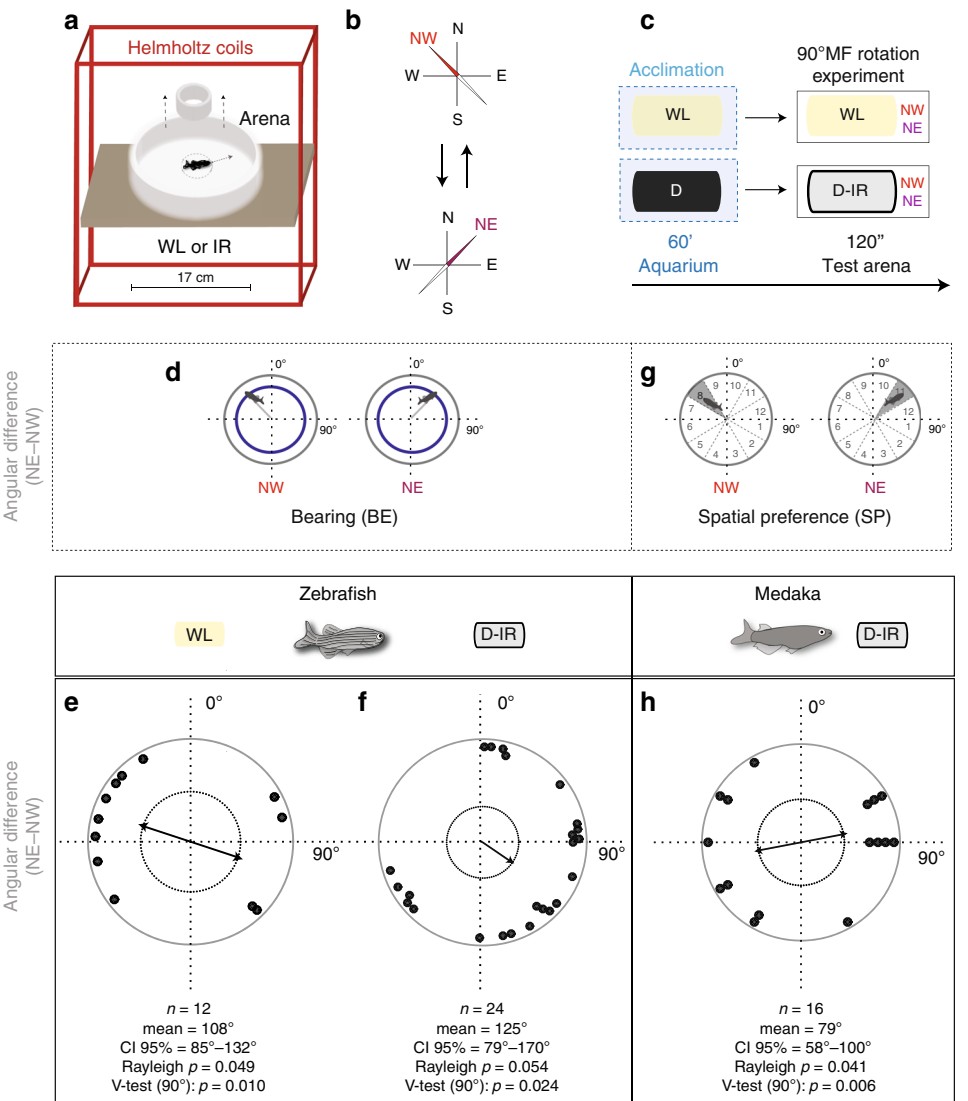

**Fig. 2** Adult zebrafish and medaka orient with respect to the direction of a magnetic field also in absence of visible light. **a–c** Schematics of the experimental setup and procedure. **a, b** In the experiment, fish are automatically released from the center of an arena and Helmholtz coils are used to deflect (change in declination) the GMF 45° towards West (NW, red) or East (NE, purple). **c** Before the experiment, fish are acclimated in white light (WL) or in darkness (D) for 60 min and tested under WL or infrared illumination (D-IR). Each fish is tested in randomized order under both conditions that differ by a 90° deflection of the MF. **d** For zebrafish, the bearing (BE) is determined as the angle from the center of the arena to the point where the fish crosses a virtual circle (radius of 6 cm). The change in preferred direction, i.e. the angular difference between the two conditions (NE−NW) is calculated for each fish. **e, f** Distribution of the angular differences for zebrafish (AB strain) in WL and D-IR. **g** For medaka, the spatial preference (SP) is assessed during the second minute after release. **h** Distribution of the angular differences for medaka (Cab strain) in D-IR. Each dot in the circular plots represents the individual angular difference, the arrow indicates the mean vector, double arrows indicate axial symmetry computed by doubling the angles. The number of fish, the mean angle with the 95% confidence interval (CI), and p values for the Rayleigh test for circular uniformity as well as the V-test (testing for circular uniformity against the alternative hypothesis of a mean angular difference of 90°) are reported. Statistical tests were performed on the axial data when such symmetry was observed

medaka IR and D-IR groups, as well the zebrafish WL group, showed only an axial trend in their directional preferences (Supplementary Fig. 9b–e).

Taken together, these data indicate that both zebrafish and medaka change their directional preference with respect to the direction of an applied MF also in the absence of visible light. The results suggest that a light-independent mechanism for magnetoreception may be a common trait of teleosts. The results do not however exclude that teleost fish might be able to employ also a light-dependent mechanism, if short-wavelength light is available.

**Young medaka change their locomotor activity in oscillating MF.** Next, we asked whether magnetosensitivity is present already in younger fish, which would be ideal subjects for subsequent studies aimed at understanding the molecular and neuronal mechanisms of magnetoreception because of their small size, transparency, and amenability to genetic modifications. We thus set up a magnetic stimulation paradigm designed to maximize the likelihood of detecting MF-dependent behavioral and neuronal responses. In this assay, each fish was allowed to explore an unfamiliar circular arena in both of the following conditions for 120 s each: static background MF (sham) and slowly oscillating

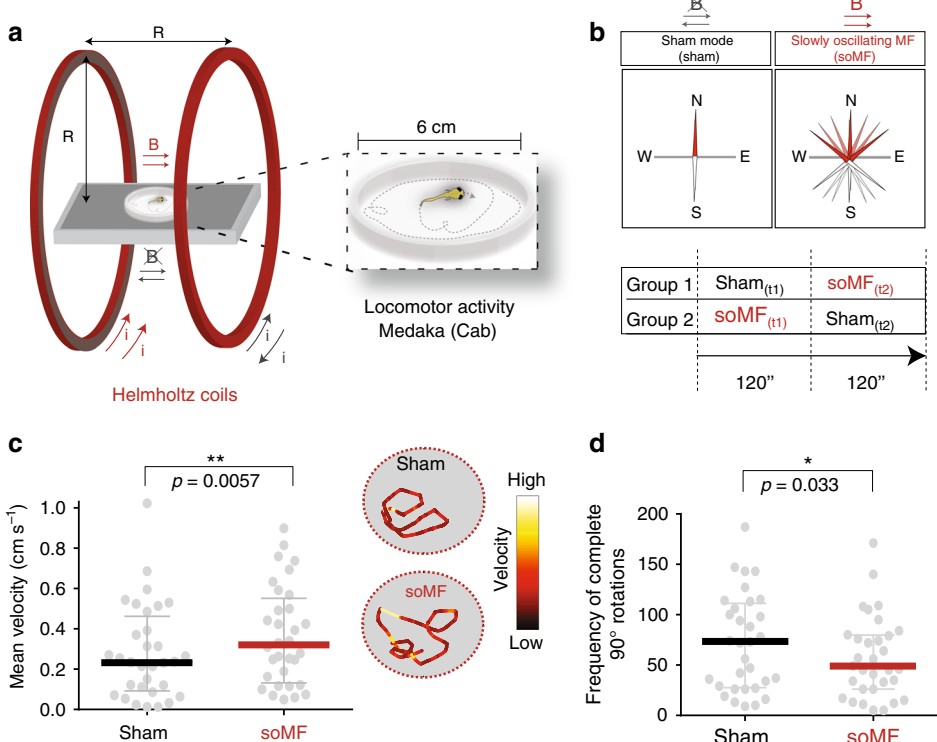

**Fig. 3** Juvenile medaka increase their locomotor activity when stimulated with a slow oscillating MF in white light. **a, b** Schematics of the experimental setup showing the test arena surrounded by double-wrapped Helmholtz coils (**a**) and the experimental conditions (**b**). Control (sham, gray): coils ran with oscillating currents in antiparallel sense resulting in no magnetic field; slowly oscillating MF (soMF, red): coils ran with parallel currents producing an electromagnetic field of 40 μT oscillating at 1 Hz in the East−West direction, resulting in continuous change of direction and intensity of the magnetic field. Juvenile fish explore a circular arena for a total of 240 s (120 s time interval ($t$) for each condition). **c, d** Graphs showing the increase of the mean swimming velocity (**c**, averaged over 60 s before and after the change to soMF), and the decrease of the frequency of complete 90° turns (**d**, counted in the 30 s interval before and after the change to soMF) in soMF as compared to the sham condition (control) for juvenile medaka (median and the interquartile range are plotted). On the right side of **c**, representative swimming trajectories of one individual during both conditions are shown color-coded by velocity. The $p$ values were derived from a two-tailed Wilcoxon signed rank test for $n = 32$ individual pairs

magnetic field (soMF), realized by applying a sinusoidally varying MF with 1 Hz frequency and 40 μT amplitude along the magnetic East−West axis with respect to the static background GMF (Fig. 3a, b). To control for non-magnetic effects such as electrical noise (Methods), we used double-wrapped Helmholtz coils in which the same amount of current is present in both conditions, either running parallel (soMF) or anti-parallel (sham)[57]. We furthermore excluded confounds due to adaptation to the arena or temporal sequence by randomizing the order of the two starting conditions (soMF and sham) for each sequential trial. Specifically, as shown in Fig. 3b, fish belonging to group 1 experienced the arena first in sham mode (sham(t1), followed by soMF(t2)), while the fish of group 2 were first tested with soMF (soMF(t1)), followed by a control phase (sham(t2)). We reasoned that if juvenile fish are indeed magnetoreceptive, a soMF that changes its direction independently of the animal's movements should generate sensory inputs in conflict with other information about the fish's body orientation (such as vestibular or proprioceptive). This should then lead to an observable change in their behavioral and neuronal responses.

In juvenile zebrafish, we did not detect any obvious differences in the swimming velocity before and after changing the magnetic condition (Supplementary Fig. 10a, b). In contrast, young medaka showed a significant increase in swimming velocity during all of the soMF phases (the mean velocity during 1 min of sham was 0.24 cm s$^{-1}$ as compared to 0.32 cm s$^{-1}$ for 1 min of soMF, Fig. 3c). An analysis of the individual experimental groups showed that the first group of fish explored the arena

with a mean velocity of 0.15 cm s$^{-1}$ (sham(t1)), which significantly increased to 0.25 cm s$^{-1}$ in the 60 s after the soMF was switched on (Supplementary Fig. 10c, group 1). The second group, which first explored the arena in soMF (soMF (t1)), also showed a significant increase in the mean velocity (0.44 cm s$^{-1}$) compared to the first group (Supplementary Fig. 10c, sham(t1)-soMF(t1)). A trend towards a decrease in velocity (0.31 cm s$^{-1}$) could also be observed after switching off soMF in group 2 (soMF(t1)-sham (t2), Supplementary Fig. 10c, group 2).

It is reported that zebrafish explore new environments by swimming spontaneously in sequences of turns that follow the same direction[58]. Therefore, we furthermore asked whether sensing the soMF would alter the structure of the exploratory behavior in zebrafish and medaka. To test this, we calculated the cumulative frequency of complete turns (90°) that occurred during exploration. While no effect was observed in zebrafish (Supplementary Fig. 10d, e), we found that during the 30 s immediately before and after the change in condition, medaka performed less complete turns in the soMF condition (Fig. 3d). When looking at group 1 and group 2 separately, a trend towards a decrease in turns could be observed in group 1 and a significant decrease between sham(t1) and soMF(t1) (Supplementary Fig. 10f).

Altogether these data suggest that juvenile medaka alter their exploratory behavior in response to a continuously changing MF, thus demonstrating that magnetosensitivity is present already in these young fish.

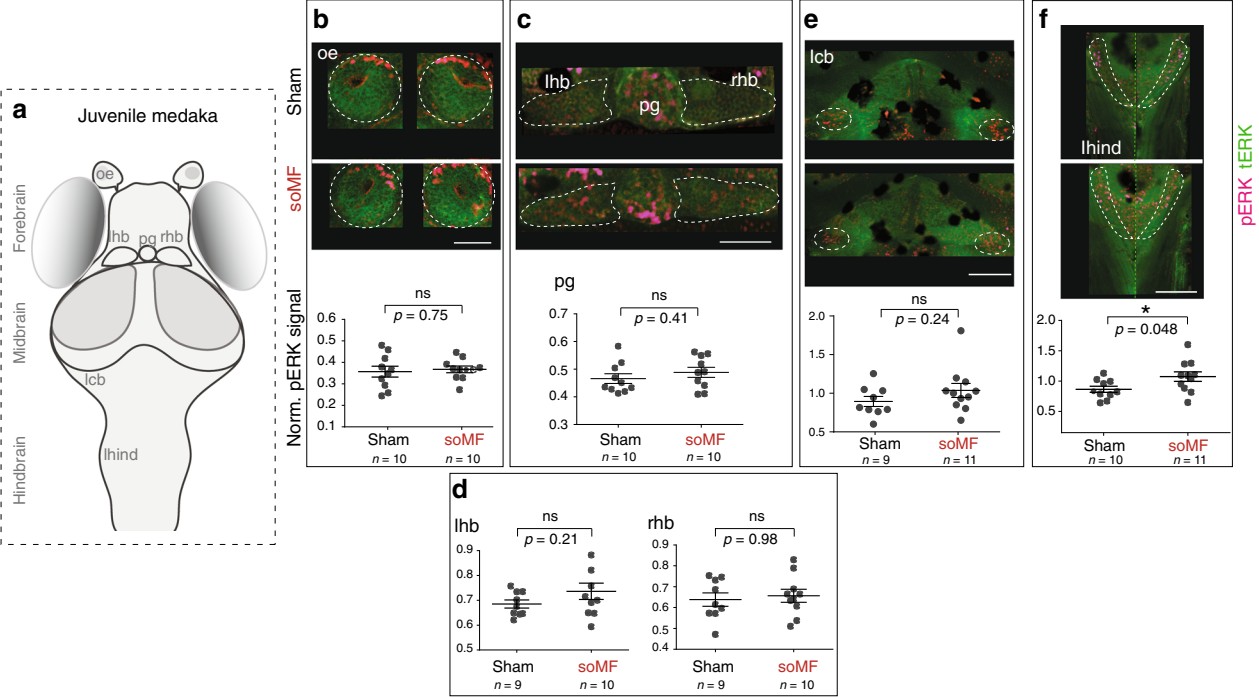

**Fig. 4** Increased number of pERK-positive neurons found in juvenile medaka hindbrain upon stimulation with a slowly oscillating magnetic field. **a** Schematic of medaka brain, dorsal view, anterior up. The major anatomical divisions into forebrain, midbrain, and hindbrain are indicated. oe olfactory epithelium, pg pineal gland, hb left habenula, rhb right habenula, lcb lateral cerebellum, lhind lateral hindbrain. **b–f** Upper panels show confocal images (maximum z-projections) of the different brain areas of interest stained against phosphorylated ERK (pERK, magenta) and total ERK (tERK, green) in juvenile medaka exposed to soMF or control condition (sham). Scale bars: 50 μm (**b, c**), 100 μm (**d, e, f**). Lower panels show the corresponding quantifications of the normalized pERK signals (pERK/tERK). The plots show mean and standard error (±SEM). A *t*-test (two-tailed) with Welch's correction for unequal variances was used and *p* values adjusted for multiple comparison by controlling the false discovery rate (FDR) at 0.05 alpha level of significance are reported. Dark spots in the images correspond to sparse pigments present in the Cab strain. Signals from regions with prominent pigmentation were not quantified

**Mapping brain regions activated by oscillating MF in medaka.** We were next interested in searching for brain areas that were differentially activated by stimulation with soMF during exploratory behavior. To this end, we chose a brain mapping technique based on immunohistochemical detection of the level of phosphorylated ERK (pERK), which shows a high degree of correlation with neuronal activity[59,60]. This approach has recently been employed to detect neural substrates of specific behaviors in freely swimming zebrafish[60].

Conveniently, whole-mount brain analyses are possible in medaka juveniles because of the small size and the relative transparency of their brains. Using the same assay as shown in Fig. 3, a cohort of juvenile medaka was stimulated for 10 min while freely exploring a test arena during soMF or with the double-wrapped Helmholtz coils run in sham mode as control (same currents but in antiparallel sense, so no resulting MF). Immediately after the experiment, fish were preserved in 4% paraformaldehyde (PFA) for subsequent immunostaining against pERK (marker of activated neurons) and tERK (total ERK, with a broad expression providing anatomical information and a reference for signal normalization[60]).

We quantified the pERK fluorescent signals in several brain regions that have previously been suggested to be involved in magnetoreception in various species, and that were readily discernible in our stained specimens (Fig. 4a). These include: (1) the olfactory epithelium (oe), suggested to contain magnetoreceptor cells in rainbow trout[61]; (2) the pineal gland (pg), (3) the habenula (hb), found responsive to MFs in rats and birds[62], the (4) lateral cerebellum (lcb) and (5) hindbrain (lhind), responsive to constantly changing MFs in homing pigeons[42,63] and European robins[39].

While our analysis of fish stimulated with soMF failed to reveal changes in pERK signal in the olfactory epithelium, pineal gland, habenulae, and cerebellum (Fig. 4b–e respectively), it showed a significant increase of activity in the lateral hindbrain (lhind, Fig. 4f). This result was replicated in an independent cohort of fish (Supplementary Fig. 11). Even though both the hindbrain and cerebellum are generally involved in controlling motor coordination and locomotion in fish[64,65], increased activation was observed only in the hindbrain. Taken together, these results identify the lateral hindbrain as a candidate brain region that may be involved in the magnetosensitivity observed in medaka.

## Discussion

In this study we adapted and developed specific behavioral assays and analyses to assess magnetoreception in two genetic teleost models, zebrafish and medaka, at different times during their life cycle. To assess magnetoreception in sexually mature fish, we chose an intra-subjective design, where each fish was tested twice under two different directions of the magnetic field . The directional preference assay that we conducted in independent cohorts of sexually mature fish (Figs. 1b and 2) showed that both species are magnetoreceptive also in the absence of visible light. However, we cannot exclude the possibility that a light-dependent mechanism also exists, one that may work in addition or in parallel whenever short-wavelength light is available. Given the evidence for the coexistence of two mechanisms that has been

reported for several species[28,30–35] (Fig. 1a), it will be interesting in the future to determine whether a light-dependent mechanism for magnetoreception is also realized in teleost fish in addition to the light-independent mechanism.

In comparison to the distribution of the angular differences calculated over the two conditions (NE–NW) in each fish (Fig. 2), we observed a larger spread when we plotted the directional preference from all initial runs in the groups (Supplementary Fig. 9) similar to an analysis previously conducted by Takebe et al.[46]. Although a larger number of observations may have revealed a stronger clustering of the directional preference of the group, as has been previously reported for zebrafish[46], our result may also be explained by the conditions in which the directional preference assay was conducted. In particular, testing fish in isolation and with no spatial cues but the MF prevents them from referencing and adjusting their individual directional choice to other spatial information and to the behavior of conspecifics, conversely to what occurs in nature within schools of fish[66,67]. Behavioral tests on cohorts of fish tested in groups and/or with additional spatial cues present may thus be informative also with respect to the ecological relevance of the observed magnetoreception for navigation purposes.

The "locomotor assay" we developed (Figs. 1b and 3) further provided evidence that already at young adult stages, medaka respond to weak and slowly oscillating MFs by changing the velocity and the structure of exploratory swimming. Interestingly, comparable hyperactivity has also recently been reported for insects exposed to oscillating MFs[7]. An increase in locomotion and feeding rate was also observed in several teleosts during natural geomagnetic disturbances[68]. With respect to young zebrafish however, the inability of our assay to detect a behavioral effect leaves it an open question whether magnetoreceptive behavior occurs later during development, or whether a different assay is needed to detect this behavior in young zebrafish.

Furthermore, the design of the behavioral assay and the small size of juvenile medaka allowed us to readily search for related brain activation patterns using whole-mount histological techniques. We identified the lateral hindbrain as a candidate region that was differentially activated during soMF-induced changes in exploratory behavior (Fig. 4f). By homology to zebrafish and other vertebrates' functional anatomy, the lateral hindbrain of medaka is likely to process inputs from cranial or peripheral sensory systems, such as the vestibular, lateral line, and trigeminal ganglia[69–71]. As discussed, an involvement of both vestibular[42,63] and trigeminal[37–40,72] systems has been proposed in vertebrates.

Live neural recordings during repeated presentations of magnetic stimuli with a systematic change of parameters, together with analyses of the connectivity patterns, will be useful to expand on the whole-mount pERK analysis performed in this study. This may help to disentangle stimulus-related neuronal activation patterns from brain activities more closely linked to magnetoreceptive behavior and to possibly also trace connections back to the candidate magnetoreceptor cells. The behavioral assays combined with neuroimaging may also provide a useful readout for forward genetic screens.

To conclude, we provide evidence for a mechanism of magnetoreception in the teleost fish zebrafish and medaka that is independent of visible light. We developed a simple assay to measure magnetoreceptive behavior already in juvenile medaka and identified the lateral hindbrain as a candidate brain region involved in magnetosensitivity using a histological brain mapping technique. These data show that the genetic and optical accessibility of these laboratory teleosts make them attractive models for in-depth follow-up studies to uncover the biophysical sensing mechanism and neuronal computation underlying magnetoreception.

## Methods

**Directional preference assay in adult zebrafish and medaka.** Adult zebrafish (AB strain) and medaka (Cab strain[73]) were used in this study (see Supplementary Table 2 for details). The fish were fed with artemia twice per day and kept in a standard 14/10 h light/dark cycle. Zebrafish were offsprings from a single parent couple and therefore genetically similar. All animal experiments were approved by the government of Upper Bavaria and were carried out in accordance with the approved guidelines. We took particular care to minimize spatial or auditory cues by installing the coil set-up in a dedicated laboratory space at the Institute of Zoology at University of Hohenheim or at the paleomagnetic laboratory "Niederlippach" of the Ludwig-Maximillian University (LMU). The test arena (Fig. 2a) was a spatially uniform glass petri dish (Ø 17 cm, H 3 cm) located inside an opaque carton box (Black hardboard TB4, Thorlabs) placed on a wooden table and covered with a black curtain (Blackout fabric BK5, Thorlabs) blocking any ambient light from reaching the test arena. In addition to the black curtain covering the arena, all other light sources in the room were switched off or covered with black tape such that all visual stimuli were abolished in the IR and D-IR conditions. The table was placed in the center of two pairs of Helmholtz coils. The coils were used to deflect the horizontal component (H) of the GMF in the room 45° towards either the East (NE) or West (NW) in a configuration in which equal amounts of current were flowing through the coil in both conditions (Supplementary Fig. 1). The coil pair in the East–West orientation generated the directional deflection of the MF, while the intensity was adjusted by applying a current in the North–South orientation (Supplementary Fig. 1), controlled with a compass and a magnetometer. The horizontal (H) and the vertical (V) component thereby remained at GMF strength ((H) 23.3 μT, V 40.5 μT, total 50.6 μT, inclination 62.6°). The environment was kept unchanged between the two conditions for each trial, except for the change in the horizontal component of the local MF. At the center of the test arena, a transparent plastic circular cylinder (Ø 6 cm) was moved up by an automatic lifting mechanism to release the fish. A custom-made IR illumination table, consisting of an array of IR LEDs (1060 nm, ELD-1060-525, Jenoptik, Germany, with a diffuser on top) was placed underneath the arena. For the WL experiments, a WL ring illumination (Leica) was centered above the petri dish. A camera sensitive over a broad range including IR was used for video recording (Sony DCR-HC23E). The IR illumination was switched on during all experiments while videos were captured at 25 frames per sec (FPS). During the acclimation phase (Fig. 2c), fish were kept individually for 1 h in small tanks placed inside boxes made of opaque material (Black posterboard TB5, Thorlabs) in either light (WL) or in darkness (D). The direction of the MF (NW or NE) was set before fish were placed individually in the inner cylinder at the center of the test arena and left there for 20 s before being released. The swimming trajectories were recorded for 1 (in the case of zebrafish) or 2 (in the case of medaka) minutes before the fish were removed from the arena and placed back to the individual box. Fish were tested randomly in the NW or NE condition and left to rest for ~45 min before being tested again in the opposite magnetic condition (or the same magnetic condition in case of the 0° experiment, Supplementary Fig. 3). Fish that were tested after acclimation in darkness were transferred to the test arena under red light illumination.

Tracking of the nose and the center of mass of the fish was performed automatically with a custom written routine in Matlab (MathWorks, MA, USA). To assess the bearing (BE) of individual fish, the program determined the point at which the nose of the fish was crossing the virtual circle of radius 6 cm (Fig. 2d). The bearing was then defined as the angle from the center of the arena to the crossing point, relative to geomagnetic North. In order to assess whether the fish did change their preferred direction between the conditions, we computed the difference in directional preference between these two magnetic conditions. The change in preference was thus defined as the angular difference in bearing between the two magnetic conditions (BE(NE) – BE(NW)) and was calculated for each individual fish. In the case where the fish were tested twice in the same magnetic condition (0° deflection, Supplementary Fig. 3), the angular difference was computed as BE(2nd trial) – BE(1st trial). Fish that did not cross the line within 1 min after release were excluded from the analysis. Manual corrections of the crossing points were made in few cases (5 of 142) in which the automated image analysis was not possible. For assessing the spatial preference (SP), the arena was divided into 12 segments and SP was defined as the segment in which the fish (using the centroid) spent most of the time during the second minute after release. SP could still be determined in case the automatic release mechanism failed. Manual corrections were made in a few cases (4 out of 87) in which the tracking software could not correctly track the fish for the entire time of the experimental run. The difference in the preferred segment between the two conditions (NE, NW) was converted to an angle (in steps of 30°) for each fish (Fig. 2g). The directional preference of the group (Supplementary Fig. 9) was assessed by analyzing the first trial of each fish normalized to the geomagnetic North. Analysis of circular statistics was performed using Oriana 4 (Kovach Computing Services). The distribution of angular differences (either polar or axial) were assessed by the Rayleigh test for clustering of data as well as compared to the expected mean of 90° by inspecting the 95% confidence interval and using the V-test[74], which tests for uniformity against an alternative hypothesis of a distribution with a specified mean (in our case 90°). The circular plots for BE were generated in Matlab by plotting one dot for each fish representing the difference angle. In the case of SP, the angular difference was binned and these values were plotted at the center of the bin. Please refer to

Supplementary Tables 2 and 3 for a summary of the conditions used in each experiment and the statistical analysis.

To analyze the thigmotaxis behavior of zebrafish (Supplementary Fig. 5, Supplementary Table 1), the zebrafish IR data were analyzed using Ethovision software (Ethovision XT, Noldus, the Netherlands). Thigmotaxis was defined as presence of the fish in a zone of the arena with 2.5 cm distance from the wall. For statistical analysis, the normality of the behavioral data was assessed in GraphPad Prism 6 (GraphPad Software, CA, USA), calculating Shapiro–Wilk normality test, KS normality test, and D'Agostino & Pearson omnibus test. Only in case the data showed a normal distribution according to at least two out of the three tests, a $t$-test was used, otherwise nonparametric tests were employed. Plots were generated in GraphPad Prism 6.

**Swimming strategy for adult medaka in light and darkness**. To assess the swimming strategy of sexually mature medaka in IR compared to WL (Supplementary Fig. 6, Supplementary Table 1) one group of fish were tested in both light conditions. Eleven fish of mixed gender of the Cab strain, aged ~6 months, were observed individually for 2 min in a circular arena (15 cm diameter). Six individuals were tested first in IR, and then in WL, while the remaining fish were tested first in WL and then in IR. The arena was illuminated from below with a ring of LEDs (1060 nm) and a WL illumination table. The IR source was always turned on, while the illumination table was switched on only for the WL trial. Whatman paper underneath the arena created a homogeneous white background. Fish were imaged for 2 min at 20 FPS from above with a near-infrared sensitive camera (Ximea MQ013RG-E2, Germany). The whole setup was placed within a black box made of carton not transparent to light (Black hardboard TB4, Thorlabs, USA).

The natural swimming behavior in the two illumination conditions were assessed with Ethovision software (Ethovision XT, Noldus, the Netherlands). Continuous looping behavior was assessed through the rotation parameter available within the software, where 720° rotations of the fish heading were counted (using 0° threshold, not accepting any backwards movement, with a minimum distance traveled of 1 cm). For statistical analysis, the normality of the behavioral data was assessed in GraphPad Prism 6, calculating Shapiro–Wilk normality test, KS normality test, and D'Agostino & Pearson omnibus test. A (paired) $t$-test was used only when the data showed a normal distribution according to at least two out of the three tests; otherwise, the nonparametric Wilcoxon signed rank test was employed. For details, please refer to Supplementary Table 1. Plots were generated in GraphPad Prism 6.

**Locomotor activity in juvenile fish**. Zebrafish of the AB strain and medaka of the Cab strain were grown at 28°C with a 14 h light/10 h dark cycle. From day 5 onwards, zebrafish larvae were raised in the fish facility under standard conditions. At 13 dpf (day post fertilization) the fish were starved for 24 h prior to the experiment. Medaka juveniles were tested within 5 days after hatching (between 10 and 12 dpf). See Supplementary Table 4 for details. A transparent circular arena (Ø 6 cm) was placed on a transparent glass table above a homogeneous LED WL source (Copic, LED Drawing light table, 22075 663, 420–700 nm), in the center of a pair of double-wrapped Helmholtz coils[23,59]. These coils work either in sham-mode or in field-mode such that the power delivered from the power supply is equal in both conditions, and thus not producing noise depending on the amount of delivered power. In the sham-oscillatory field condition (control) the currents were also oscillatory at the same frequency as in the experimental condition. An MF was generated when the current ran in parallel, while in sham mode the currents ran in opposite directions within the coils resulting in no net MF at the test arena while producing the same electric noise and ohmic heating (which was however minimal). An oscillating MF (sinusoidal with peak value 40 µT oscillating at 1 Hz) was applied in the East–West direction (Fig. 3), resulting in a continuous change with both direction and intensity of the field. The swimming of the fish in the arena was imaged with the Ximea MQ003MG-CM camera from above (the custom-made setup is depicted in Supplementary Fig. 12). The fish were introduced in the arena containing fresh water (same as the tank water in which the two species are raised and cultured) with the oscillating field either in the magnetic (soMF) or in sham mode (sham). After 2 min, the stimulation was changed to soMF (group 1) or sham (group 2), respectively, and the fish were observed for another 2 min.

Analysis of the locomotor activity was performed using Ethovision software (Ethovision XT, Noldus, the Netherlands). Velocity and number of turns were computed automatically and in the case manual corrections to the swimming trajectories were necessary, this was done without knowledge of the experimental condition. Turns were defined as 90° rotations of the fish heading (using 0° threshold, not accepting any backwards movement). Fish that were inactive (mean swimming velocity below 0.01 cm s$^{-1}$) were excluded from the analysis. Behavioral data were tested for whether they were consistent with a normal distribution using three tests for normality (Shapiro–Wilk, KS, and D'Agostino & Pearson omnibus normality test as implemented in GraphPad Prism 6). If at least two out of three tests were consistent with a normal distribution, a $t$-test was used for statistical analysis. Otherwise, nonparametric tests were employed, respectively. For details, refer to Supplementary Tables 4, 5, and 6 for a summary of the statistical analysis. Plots were generated in GraphPad Prism 6.

**Mapping of neuronal activation after stimulation with soMF**. Using the same setup as in Fig. 3a and as explained above, juvenile medaka were stimulated while swimming together as a group either with MFs oscillating at 1 Hz at the intensity of the GMF (soMF condition) or in sham mode (control condition). After 10 min, fish were collected and quickly preserved in 4% PFA (Sigma Aldrich) in PBS + 0.25% Triton (PBTr) overnight at 4°C. The juvenile fish were then immunostained using a protocol similar to the one described in Randlett et al.[63]. Briefly, the samples were placed in 2 ml Eppendorf tubes and washed three times for 15 min in PBTr at room temperature, shaking at a speed >400 rpm. Next, the samples were treated once for 5 min with 150 mM Tris-HCL pH9 at room temperature and then incubated once for 15 min at 70°C. The samples were washed as above, once for 5 min in PBTr and were subsequently incubated with-Trypsin EDTA (Life Technology, diluted 1:20 in PBTr) for 45 min on ice. Five washes were performed as above, each for 5 min. The samples were then incubated in Blocking Buffer (5% goat serum, 1% BSA, 1% DMSO in PBTr) for 1 h at room temperature, continuously shaking at a speed >400 rpm. Subsequently, the samples were incubated in primary antibodies (anti-pERK from Cell Signaling, 4370 and tERK from Cell Signaling, 4696) at the dilution of 1:600 in Blocking Buffer, shaking over two nights at 4°C. After washing ten times, each for 5 min as above, samples were incubated in secondary antibodies (anti-rabbit DyLight 594 from ThermoFisher Scientific and anti-mouse AF488 from Abcam, ab150113), at a dilution of 1:500 in Blocking Buffer shaking over two nights at 4°C. The samples were then washed ten times, each of 5 min and mounted on slides in 80% glycerol (in PBS). Samples were subsequently imaged using confocal microscope (Olympus FLUOVIEW FV12, ×20 objective), using a 559 nm laser (to detect pERK signal) and 488 nm laser (to detect tERK signal). The same laser intensity (15%) was used for all confocal scans. Each confocal scan was performed using a voxel size of $0.62 \times 0.62 \times 2.5$ µm ($x \times y \times z$). Individuals stimulated with soMF and controls were imaged in a random sequence.

All data processing was performed by two scientists with the group assignments of the data sets encrypted, using Fiji[75] and custom Matlab routines. To quantify neuronal activation, the mean intensity of the pERK signal in each brain region was computed for each of the selected z-planes. The pERK signal was normalized dividing it by the mean intensity of the tERK signal for each plane[63]. Statistics were computed in GraphPad Prism 6 and in R[76]. Shapiro–Wilk normality test, KS normality test, and D'Agostino & Pearson omnibus test were employed to assess the normality of the data. If data were normally distributed (based on the results from at least two out of three of the above-mentioned tests), they were analyzed using a two-tailed Welch's $t$-test (see Supplementary Table 7 for details). The graphs display the mean and the standard error of the mean (±SEM) and were generated with GraphPad Prism 6. For display of example results in Fig. 4, maximum z-projections were computed in Fiji[75].

**Data availability**. Data are available upon reasonable request to the corresponding author.

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

## Acknowledgements

We thank Dr. Baubak Bajoghli for sharing medaka Cab strain, Dr. Sabrautzki for her support and for her valuable feedback on the animal experiments, Anja Stelzl for contributing to animal husbandry and raising the juveniles, Hannes Rolbieski for administrative support, Panagiotis Symvoulidis for feedback on the pERK analysis, Christiane Fuchs and Hannah Busen for statistical advice, Susanne Seitz and Bahar Najafi for access to microscopy, and Chris Penningroth for careful reading of the manuscript. S.H.K.E acknowledges support from DFG grant Ed258/1-1. M.W. acknowledges funding from the Human Frontier Science Program (RGP13/2013). A.M., A.L., and G.G.W. are grateful for support from the European Research Council under grant agreement ERC-StG: 311552 awarded to G.G.W.

## Author contributions

A.M. and A.L. designed, performed, and analyzed all the experiments. S.H.K.E and D.S. designed and built the coil systems for the experiments in sexually mature fish and performed preliminary experiments. S.H.K.E. contributed to the design of the experiments conducted on juvenile fish and built the double-wrapped Helmholtz coils used for this experiment, performed the experiments on the sexually mature fish together with A. M. and A.L. and gave feedback on the manuscript. M.C. wrote the algorithm used for automated tracking and analysis of the bearing of the adult fish and gave feedback on the statistical analysis. W.W. supported the project and provided feedback. M.W. contributed to the design of the adult experiments, supported the project and provided detailed feedback on the manuscript. A.L. conceived and generated the illustrations. G.G.W. designed, coordinated, and supervised the study. A.L., A.M., and G.G.W. wrote the manuscript.

## Additional information

**Competing interests:** The authors declare no competing financial interests.

