## [Peer Review File · Nature Communications]

Reviewers' comments:

Reviewer #1 (Remarks to the Author):

In this manuscript, Myklatun, Lauri, et al. present a provocative study, where they present evidence that adult zebrafish and adult Medaka respond behaviourally to changes in the orientation of the magnetic field independent of visible light. Importantly, these experiments are performed at magnetic field strengths equal to the geomagnetic field, indicating that these fishes can sense and could use the orientation of the earth's magnetic field as a behavioural cue. Additionally, they performed experiments on larval stage fish, and find that an oscillating magnetic field induces hyperactivity in medaka, but interestingly not in larval zebrafish, indicating a developmental time course for magnetoreception in zebrafish. Finally, they use pERK as a marker of active neurons in the larval medaka brain, and conclude that the oscillating magnetic field induces activity in a lateralized pattern in the hindbrain.

This is an interesting study, but I remain skeptical and have some significant concerns with the experimental design and analyses. While I am not an expert specifically in magnetoreception/navigation and its typical analyses, I believe these are critical considerations which should be dealt with prior to publication.

My concerns are related to

1) The analysis of adult reorientation behaviour

I find the reorientation analysis performed to be confusing and difficult to understand, perhaps for a researcher outside of this specific discipline like myself. But, as I understand it, the fish are first released into a visibly homogenous environment with a given magnetic field orientation, and an analysis of the initial bearing direction of the fish (or its preference for a particular quadrant of the environment), is scored. This experiment is then repeated, with the orientation of the magnetic field being rotated by 90 degrees. They then take the difference in angle between these two trials, and ask if this change is consistent with a 90 degree rotation. They find this to be the case in most experiments. However, if the null hypothesis is that the fish are not responding to the magnetic field, and assuming a perfectly homogenous environment lacking spatial cues, the bearing orientation would be assumed to be random in each trial. Therefore, some amount of "reorientation" would be expected, how much of which might depend on how homogeneous the environment actually is. I feel a critical control would be to directly compare the distributions of reorientation when fish are tested with paired trials under the same magnetic field vs paired trials where the magnetic field is rotated. If these reorientation distributions are clearly different, then I believe this would be much better evidence for a response to the reoriented magnetic field.

The data shown and the tests performed are all done on the change in angle between the rotated conditions. From this I am assuming that the actual orientation of the fish is not significantly biased by the magnetic field at a population level (if this is not the case, then this should be shown). Therefore, the claim seems to be that each individual fish has a preference for its individual orientation within the magnetic field, but this is random with respect to its conspecifics. Of course it is possible that such a system could exist, but what utility could it serve in an evolutionary or navigational context? This should be discussed.

2) pERK analysis

The authors compare the number pERK stained neurons in larval medaka either stimulated with an oscillating magnetic field or a stationary one. They then search throughout the brain, reporting on

statistical tests for 30 different analyses. Different quantifications are performed in different areas, either counting observable cells, the mean value of the area normalized to tERK, etc. It is a bit concerning that the analysis is not consistent and therefore not equivalent. This is explained in the text as some analyses were not possible in some areas, but why can't mean of the area be used in all regions?

They determine that two analyses of the lateral hindbrain (ratio left/right, control-OMF) and (left-left, control-OMF) show 'significant' differences based on t-tests, with the relatively modest p values of $p = 0.043$ and $p = 0.048$ respectively. This would appear to indicate that the left lateral hindbrain is specifically activated by the oscillating magnetic field. This difference is not obvious in the chosen example images (4b), indicating that if there is an effect, it is fairly subtle. Since the authors perform multiple analyses throughout different areas of the same brains, I believe this suffers from the "multiple comparisons problem" and therefore any conclusion drawn from the $p < 0.05$ result is not statistically sound. If one test is performed, then an alpha value of 0.05 is thought of as acceptable for null hypothesis rejection. However, since 30 tests are performed, the family wise alpha rate is something closer to $(\alpha_T = 1 - (1 - \alpha)^{n_{\text{Tests}}}) = 0.78$. Therefore, it is quite likely to see one or two false positive results using a 0.05 alpha rate on any individual test, even if no real difference exists in the two populations. Proper multiple comparisons controls, such as the Bonferroni correction would set the threshold alpha value at 0.0017, well below the observed values. If the authors want to claim there is a real effect of lateralized activation in the hindbrain, I believe these experiments must be duplicated in a second biological replicate targeted specifically to the lateral hindbrain.

3) Oscillating magnetic field experiments on larval medaka.

The authors explain that the dual-wrapped design of their coils ensures that a constant amount of current is applied at all times, and thus the same amount of electrical noise. However, it is really the case that there is no audible difference between the oscillating magnetic field condition and the constant condition? One might imagine a change in frequency, which could have an influence on the behaviour of the fish. Can the authors record the resultant noise/vibrations at the dish, and compare between the two conditions? If these differ, then a control addressing the possibility that the fish can hear the oscillation patterns in the coils needs to be addressed.

Reviewer #2 (Remarks to the Author):

This is an interesting study testing magnetic orientation in two model species of fish in behavioural experiments and examining brain activity in response to magnetic fields. The study is well conceived, though I disagree with the title and conclusions drawn from the results, i.e. the statement that magnetoreception is light independent. The data show that the fish respond to a shift of the magnetic field under IR and your (undefined) white light condition (see comments about spectral properties of light below). I agree that these results show that the fish use a light-independent mechanism to orient under IR, and that they likely use the same mechanism under the WL condition. However, this does not mean that they do not also have a light-dependent magnetic compass, as has been shown in newts and birds, which use both systems under different ambient conditions and in different behavioural contexts (see work by Wiltschko, Phillips, Muheim). Birds, for example, if tested under high-intensity light of different spectra, can become disoriented or revert to light-independent magnetic alignment responses. Also, depending on the behavioural context, animals might not use a light-dependent compass, but show an alignment behavior because they try to read magnetic map information. This could be the case also here, so all you can say is that under the experimental conditions that you used and in the behavioural context you tested the fish they used a light-independent magnetic sense. Given the broad phylogenetic occurrence of radical-pair-based

magnetoreception in the animal (and plant) kingdom, and that fish have six cryptochrome genes, I would be very careful in suggesting that they do not also have a light-dependent magnetic compass, in addition to light-independent magnetoreception that you describe here.

Specific comments:

Analysis of bearing and spatial preference:

I don't understand why the data is binned into 8 sectors, when to start with, you determined both bearing and spatial preference to a precision of one degree. By binning your data, you lose potentially important information, thus I see no advantage of doing so. Also, why do you bin your data in 45 deg sections starting at 0 deg, instead of $0 \text{ deg} \pm 12.5 \text{ deg}$? Figs. 2d-f are very difficult to interpret, since 90 deg is not the center of a bin, which I think is very odd, given that you test your data against the expected 90 deg change. Thus, I recommend to (1) not bin the data at all, or (2) center the data so that 90 deg is the center of one bin (as you appear to have done with spatial preference in Figs. 2g+h).

Use of V-test: I don't think that you can use the V-test in this context, since your "bearings" are differences between two tests. The authors should carefully check whether they can use a V-test. The alternative would be to use the confidence interval test, as described by Batschelet.

The difference between the NW and NE condition was calculated by subtracting BE(NE) from BE(NW) = $BE(NE) - BE(NW)$, but you tested about half of the animals first under NW and the other first under NE. If true, your differences are incorrect, since the difference in orientation in fish first tested under NE and then NW should be calculated $BE(NW) - BE(NE)$. This might explain the axiality in your data, so please check!

Use of term "GMF": It is misleading to use the abbreviation for geomagnetic field for artificially deflected fields. Even though they have the same properties as the geomagnetic field, they are artificially created magnetic fields. It is important to distinguish whether, for example, the natural GMF was used as control and an artificially changed field as the experimental field, or whether both control and experimental fields were created with coils. If you use the term GMF you suggest that you used the Earth's magnetic field, which is not true, so please use another term. Please also give the properties (total intensity, inclination) of your local geomagnetic field.

Use of term "oMF": Also this term can be misunderstood since some research groups (e.g., Mouritsen lab) use OMF as abbreviation for oscillating RF-fields, which is a different type of oscillating field in regard of its effect on a magnetic sense. If I understand you correctly, your field is a static magnetic field oscillating in intensity and direction at 1 Hz, while an RF-field is a much weaker magnetic field (tens of nT) oscillating in the MHz frequency along in direction. Thus, your field is a means to test for general sensitivity of animals to magnetic fields, while RF-fields specifically test the radical-pair mechanism. So, even though the hyperactivity in your fish and the one in cockroaches are due to the oscillating fields, the underlying reasons are very different because of the different properties of the two oscillating fields. So, it is very important to make this distinction!

Specification of white light (WL): please provide spectral information, incl. irradiance of your "white light". The current description does not provide any useful information. How is the light produced (LED, iridescent, halogen), does it include UV, how does white balance look like (short- vs. long-wavelength light)? Work from birds and newts have shown that spectral composition and light intensity are important properties in magnetic compass orientation, thus depending on the properties of the white light used, radical-pair-based magnetoreception might not have been possible for the fish to use, and could be the reason that you did not find light-dependent magnetoreception (see work by Wiltschko, Phillips, Muheim).

Reviewer #3 (Remarks to the Author):

John Phillips

Review for Nature Communications: Myklatun et al. 'Magnetoreception in teleosts is light-independent and influences their locomotor and neural activity.'

The authors report important advances in the development of a new assay to explore the molecular/genetic underpinning of the magnetic sense in teleost fish. There are a number of interesting findings, and considerable potential for larval fish systems to contribute to a better understanding of this novel sensory systems(s). However, there are several issues with the summary of earlier literature, and more importantly with aspects of the statistical analyzes, that need to be addressed.

Background: With respect to the literature on magnetoreception, several issues need to be clarified. First, the papers cited in Figure 1 should be included in the literature cited. These papers, as well as papers cited elsewhere in the manuscript, are a rather odd selection from the literature. There are no papers cited from the large literature on light-dependent magnetic compass orientation in birds and amphibians, with the exception of the paper by Stapput et al. (2006) that reports magnetic alignment responses in the dark by a migratory bird. What is missing, both in the papers that the authors' cite and in their discussion of the literature, is the large body of evidence indicating that both amphibians and birds have two distinct magnetoreception mechanisms (Phillips 1986, Munro et al. 1997) --- a light-dependent, axial/inclination magnetic compass (Phillips & Borland 1992, Wiltschko et al. 1993, Deutschlander et al. 1999) and a non-light-dependent, polarity sensitive 'map' detector involved in sensing subtle spatial variation in the magnetic field used to derive a map or geographic position sense (Phillips & Borland 1994, Munro et al. 1997, Phillips et al. 2002). By only citing the paper by Stapput et al. from the long list of behavioral experiments carried out in the Wiltschko lab (the other citations for birds in Fig 1 are for neurophysiological experiments), the authors give the impression that birds have only a non-light-dependent magnetoreception mechanism, which is clearly not the case (Wiltschko et al. 2010).

Likewise, the authors' citations of behavioral evidence in rodents in Figure 1 are primarily from molerats tested in the dark, and a somewhat offbeat paper by Prato et al. The finding that molerats can orient in the dark, and appear to have a non-light-dependent magnetoreception mechanism, is not surprising given that these rodents are adapted to live in aphotic subterranean habitats. There is no mention of neurophysiological/biochemical evidence for a light-dependent magnetoreception mechanism in the retinas of murine rodents, or of behavioral evidence for a well-developed magnetic compass that is sensitive to low-level radio frequency fields (Deutschlander et al. 2003, Olcese 1990, Muheim et al. 2006, Phillips et al. 2013, Malkemper et al. 2015).

Given the presence of both magnetite-based magnetoreceptors in molerats and light-dependent, RF-sensitive magnetoreceptors in murine rodents, it would not be surprising if murine rodents, and perhaps other mammals, also have dual mechanisms, and that this might also be the case in fish. To date there has been no attempt to investigate whether murine rodents or teleosts have a magnetite-based magnetoreception mechanism (e.g.) under conditions where input from an RF-sensitive, light-dependent magnetic compass would be eliminated or downgraded.

A more thorough discussion of the literature is needed because the findings reported by the authors point to the presence of a non-light-dependent magnetoreception mechanism in teleosts, but provide no evidence about whether a second, light-dependent mechanism is also present. This is important because the neural activity studies of larval medaka were carried out under white light. Therefore, although the authors present evidence for the involvement of a non-light-dependent magnetoreception mechanism in mediating some of the behavioral responses of adult fish and larval medaka, their findings do not distinguish between the possibilities that the increased neural activity in larval medaka is due to a light-dependent mechanism, a non-light-dependent mechanism, or both.

Should the authors choose to carry out future research to distinguish between these alternative possibilities (which is not necessary for the current manuscript), they should keep in mind that electromagnetic shielding should be used to screen out low-level radio frequency interference (e.g., radiated by video cameras, stimulus coils, microscope lights, as well as computers and other equipment located nearby) that has been shown to increase the variability of responses mediated by putative RPMs (Muheim et al. 2006, Vacha et al. 2009, Phillips et al. 2013, Engles et al. 2014, Landler et al. 2015, Malkemper et al. 2015).

There are several issues with the data and statistical analysis that should be addressed. The 'V-test' is no longer widely used in the spatial literature. This is because the test statistic (the component of the mean vector that is in the expected direction) is dependent both on the mean vector bearing and mean vector length—i.e., for a given sample size, the same level of significance can be obtained from a scattered distribution with a (relatively weak) mean vector that coincides with the expected direction and a tightly clustered distribution with a very strong mean vector with a large deviation (up to 70-80 degrees) from the expected direction, i.e., both could have the same component in the expected direction. There was a lengthy discussion of this in the literature some years ago, and the consensus was that a Rayleigh test should be used to determine if there is greater clustering in the distribution of bearings than would be expected by chance. If so, a 95% confidence interval for the mean vector bearing can then be used to determine if the mean vector bearing is consistent with the expected direction. In the case of a tightly clustered distribution of bearings with a mean vector that deviates substantially (e.g., by 40-50 degrees or more, but less than 90 degrees) from the expected direction, a possible conclusion is that there is strong clustering in the distribution of bearings, but that this orientation deviates significantly from the expected direction.

With respect to the distributions showing axially bimodal responses. By plotting each data point twice ('double plotting'), a practice that has been used occasionally in the literature, the authors indicate that the direction of the each response along the 'axis of response' is unimportant. However, this needs to be clearly stated in the methods and/or figure caption; my apologies if it was stated somewhere in the manuscript and I missed it.

Each of the distributions should be plotted with only one data point for each bearing (regardless of whether the bearings are unimodally or bimodally distributed) to help avoid confusing readers who might find a disconnect between the sample size and the number of bearings when the bearings for the axially bimodal distributions are 'double plotted' (e.g., Fig 2d,e; S3a,d; S6a). Notice also that in Fig S6, the number of bearings is greater than the sample size given, although in this case the discrepancy is not due to double-plotting. Also in Fig 2h, the sample size agrees with the number of bearings, but the distribution of bearings is symmetrical (consistent with double plotting) so might be worth double-checking.

Once the numbers of bearings vs. samples sizes have been corrected in the axially symmetrical distributions, it is very important that the authors show a figure of the actual distribution of bearings in each figure, i.e., without double plotting. For distributions that are axially symmetrical, a second distribution can be included with the bearings doubled (each 'doubled' bearing plotted only once) and a unimodal mean vector bearing calculated from the distribution of doubled bearings (Note: this is not the same as double-plotted bearings) to show if the bimodal distribution is non-randomly distributed. It's important to emphasize yet again the importance of plotting the distribution of individual bearings for each experimental treatment (i.e., whether or not the distribution shows axial symmetry), so readers can accurately assess the sample sizes, and visually compare the distributions under different experimental treatments. I would also encourage the authors to use a two-sample test to compare the distributions of doubled bearings in the different treatment conditions, e.g., with a two-sample test (e.g., Watson U2). It is quite possible that the responses of adult zebrafish under WL are significantly different from those under IR and D-IR, which may turn out to be interesting given the evidence that other vertebrates have dual light-dependent and non-light-dependent magnetoreception mechanisms (see above). There may also be a difference between the responses of adult medaka under IR and D-IR. Needless to say, interpretation of the patterns of neural activity depends not only

on their being an effect, but also on whether or not there are multiple effects and, perhaps, multiple magnetoreception mechanisms involved.

One final point of information—the authors mention that some populations of medaka migrate back and forth between marine and freshwater habitats. I assumed that larval zebrafish and medaka were both tested in fresh water, but couldn't find this stated anywhere in the manuscript. If the larval medaka (but obviously not larval zebrafish) were tested in salt or brackish water, an additional control(s) should be carried out to make sure that the larvae were not responding to an electrical field induced by the 1 Hz oscillating magnetic field.

Please feel free to contact me by e-mail (Phillip@vt.edu) to discuss any or all of these issues.

John Phillips

Literature cited

Deutschlander, M.E., S.C. Borland & J.B. Phillips 1999. Extraocular magnetic compass in newts. *Nature* 400: 324-325.

Deutschlander, M.E., M.J. Freake, S.C. Borland, J.B. Phillips, L.E. Anderson & B.W. Wilson 2003. Learned magnetic compass orientation by the Siberian hamster, *Phodopus sungorus*. *Anim. Behav.* 65: 779-786.

Engels, S. et al. 2014. Anthropogenic electromagnetic noise disrupts magnetic compass orientation in a migratory bird. *Nature*

Landler, L., Painter, M.S., Youmans, P.W., Hopkins, W.A., Phillips, J.B. 2015. Radio frequency field affects association of magnetic field with novel surroundings in yearling snapping turtles (*Chelydra serpentina*). *PLoS ONE* 10(5): e0124728. doi:10.1371/journal.pone.0124728.

Malkemper, E. P., S. H. K. Eder, S. Begall, J. B. Phillips, M. Winklhofer, V. Hart & H. Burda 2015. Magnetoreception in the wood mouse (*Apodemus sylvaticus*): influence of weak frequency-modulated radio frequency fields. *Sci. Rep.*, 9917; doi:10.1038/srep09917.

Muheim, R, N.M. Edgar, K.S. Sloan & J.B. Phillips 2006. Magnetic compass orientation in C57BL6 mice. *Learn. Behav.* 34: 366-373.

Munro, U., J.A. Munro, J.B. Phillips, R. Wiltschko & W. Wiltschko 1997. Evidence for a magnetite based navigational 'map' in birds. *Naturwissenschaften* 84:26-28.

Olcese, J. M. The neurobiology of magnetic field detection in rodents. *Prog Neurobiol* 35, 325–330 (1990).

Phillips, J.B.. 1986. Two magnetoreception pathways in a migratory salamander. *Science* 233:765 767

Phillips, J.B. and S.C. Borland. 1992. Behavioral evidence for the use of a light dependent magnetoreception mechanism by a vertebrate. *Nature* 359:142 144

Phillips, J.B. and S.C. Borland. 1992. Magnetic compass orientation is eliminated under near infrared light in the eastern red spotted newt *Notophthalmus viridescens*. *Anim. Behav.* 44:796 797

Phillips, J.B. and S.C. Borland. 1994. Use of a specialized magnetoreception system for homing. *J. Exp. Biol.* 188:275 291

Phillips, J.B., K.A. Adler and S.C. Borland. 1995. Navigation by an amphibian. *Anim. Behav.* 50: 855 858.

Phillips, J.B., M.J. Freake, J.H. Fischer, S.C. Borland 2002. Behavioral titration of a magnetic map coordinate. *J. Comp. Physiol.* 188: 157-160.

Phillips, J.B, S.C. Borland, M. J. Freake, J. Brassart & J.L. Kirschvink 2002. "Fixed Axis" Magnetic Orientation by an Amphibian: Non Shoreward Oriented Compass Orientation, Misdirected Homing, or Positioning of a Map Detector in a Consistent Alignment Relative to the Magnetic Field? *J. Exp. Biol.* 205: 3903-3914.

Phillips, J.B., P.W. Youmans, R.Muheim, K. Sloan, L Landler, M.S. Painter, & C. Anderson 2013. Rapid learning of magnetic compass direction by C57BL/6 mice in a "plus" water maze. *PLOS ONE* 8: e73112.

Ritz, T., P. Thalau, J.B. Phillips, R. Wiltschko & W. Wiltschko 2004. Avian Magnetic Compass: Resonance Effects Indicate a Radical Pair Mechanism. *Nature* 429: 177-180.

Stapput, K., P. Thalau, R. Wiltschko & Wiltschko, W. 2006. Orientation of birds in total darkness. *Cur. Biol.* 18: 602-606.

Wiltschko, W., Munro, U., Ford, H. & Wiltschko, R. 1993 Red light disrupts magnetic orientation of migratory birds. *Nature* 364, 525-527.

Wiltschko, R., Stapput, K., Thalau, P. & Wiltschko, W. 2010. Directional orientation of birds by the magnetic field under different light conditions. *J. R. Soc. Interface* 7, S163-77

Vacha, M., Puzova, T. & Kvcialova, M. Radio frequency magnetic fields disrupt magnetoreception in American cockroach. *J Exp Biol* 212, 3473-3477 (2009).

See also:

Phillips, J. B., P.E. Jorge, & R. Muheim 2010. Light-dependent magnetic compass orientation in amphibians and insects: candidate physiological and molecular mechanisms *J. Roy. Soc. Interface* 7: S241-256.

Phillips, J.B., R. Muheim, & P. E. Jorge 2010. A behavioral perspective on the biophysics of the light-dependent magnetic compass: a link between directional and spatial perception? *J. Exp. Biol.* 213: 3247-3255.

General Reply to all Reviewers:

We thank all reviewers for the constructive feedback which helped us tremendously to improve the manuscript. As a consequence of the combined comments, we have updated the statistical analysis of the behavioral data on adult preference by computing the Rayleigh test with confidence intervals in addition to the more powerful V-test. Additionally, we now also provide the distributions of the cohort directional preferences.

As requested, we have also replicated the main finding from the brain mapping study (revised Fig. 4) in the juvenile medaka in an independent cohort of animals, which also showed increased activity in the lateral hindbrain under a slow oscillating magnetic field (soMF).

We have furthermore adapted the title and text as requested to clarify that we did not intend to imply that a light-independent magnetoreceptive sense in zebrafish and medaka excludes the presence of also a magnetic sense that operates with visible light in the same fish. We now discuss the possibility of a dual mechanism in depth, as requested by reviewers 2 and 3.

We thank the reviewers again for their valuable points that greatly helped to improve our work. Please find our point-by-point answers below, cross-referenced to the updated figures, data tables, as well as edits in the main text and supplementary information.

Reviewer #1 (Remarks to the Author):

In this manuscript, Myklatun, Lauri, et al. present a provocative study, where they present evidence that adult zebrafish and adult Medaka respond behaviourally to changes in the orientation of the magnetic field independent of visible light. Importantly, these experiments are performed at magnetic field strengths equal to the geomagnetic field, indicating that these fishes can sense and could use the orientation of the earth's magnetic field as a behavioural cue. Additionally, they performed experiments on larval stage fish, and find that an oscillating magnetic field induces hyperactivity in medaka, but interestingly not in larval zebrafish, indicating a developmental time course for magnetoreception in zebrafish. Finally, they use pERK as a marker of active neurons in the larval medaka brain, and conclude that the oscillating magnetic field induces activity in a lateralized pattern in the hindbrain.

This is an interesting study, but I remain skeptical and have some significant concerns with the experimental design and analyses. While I am not an expert specifically in magnetoreception/navigation and its typical analyses, I believe these are critical considerations which should be dealt with prior to publication.

Reply: We thank the reviewer for recognizing the value of our work on magnetoreception in zebrafish and medaka. We are grateful for the comments and addressed the concerns in detail below.

My concerns are related to

1) The analysis of adult reorientation behaviour

I find the reorientation analysis performed to be confusing and difficult to understand, perhaps for a researcher outside of this specific discipline like myself. But, as I understand it, the fish are first released into a visibly homogenous environment with a given magnetic field orientation, and an analysis of the initial bearing direction of the fish (or its preference for a particular quadrant of the environment), is

scored. This experiment is then repeated, with the orientation of the magnetic field being rotated by 90 degrees. They then take the difference in angle between these two trials, and ask if this change is consistent with a 90 degree rotation. They find this to be the case in most experiments. However, if the null hypothesis is that the fish are not responding to the magnetic field, and assuming a perfectly homogenous environment lacking spatial cues, the bearing orientation would be assumed to be random in each trial. Therefore, some amount of "reorientation" would be expected, how much of which might depend on how homogeneous the environment actually is. I feel a critical control would be to directly compare the distributions of reorientation when fish are tested with paired trials under the same magnetic field vs paired trials where the magnetic field is rotated. If these reorientation distributions are clearly different, then I believe this would be much better evidence for a response to the reoriented magnetic field.

Reply: Although the reviewer fully understood the experimental design, we apologize that we have not explained the design clearly enough. We have now improved on it in the main text and Methods section, by explaining the following aspects in more detail (pg. 3/7-8 and revised Fig. 2).

We took particular care to minimize spatial or auditory cues by installing the coil set-up in a dedicated laboratory space for paleomagnetic measurements in a remote environment 80 km outside of Munich. The experimental arena (circular dish) was spatially uniform and was surrounded with a thick black curtain, such that all visual stimuli were abolished in the IR and D-IR conditions. The environment was kept unchanged between the two conditions for each trial, except for the change in the horizontal component of the local magnetic field. This was deflected 45° towards East or West in order to achieve a symmetric design with the same amount of current running in the coils (just in opposite direction) in both conditions. The intrasubjective design with a total deflection of 90° of the magnetic field between repeated trials minimized a possible effect of stationary spatial cues and tested for a change of orientation in each fish. To further avoid biases, the fish were released from the center by an automated mechanism in all trials. Taken together, the experimental design made a clear prediction of a change in the directional preference consistent with the rotation of the magnetic field by 90°.

In order to assess whether the fish did indeed change their preferred direction with respect to the direction of the magnetic field, we computed the difference in bearings between these two magnetic conditions. We observed a distribution of these difference angles consistent with the rotation of the magnetic field in zebrafish under white light (WL) and in zebrafish and medaka under infrared (IR) illumination after acclimation to darkness (D-IR condition), also with the modified analysis requested by the reviewers 2 and 3 (including the Rayleigh test in combination with confidence intervals of the mean, revised Fig. 2). The IR-group of both species did not show a significantly clustered distribution of the angular differences by the Rayleigh test. These data are now in the revised Fig. S4.

We thank the reviewer for suggesting an additional control experiment with a 0° change of the magnetic field applied between the two trials, *i.e.* testing the fish twice in the same magnetic field condition. We obtained permission to use the dedicated laboratory space of the university again and performed this experiment with a cohort of zebrafish derived from one parent couple and of similar age as the previous cohorts.

We ran the experiment under IR illumination after acclimation in darkness for 60 minutes (D-IR condition) to avoid visual stimuli. We furthermore tested half of the group twice in NE while the other half of the group was tested twice in NW, in a randomized order for each pair of trials. This allowed us to exclude an absolute preference in the arena independent of the magnetic field direction (Rayleigh test: $p = 0.98$).

We calculated the difference between the two trials as 2nd trial- 1st trial, and observed a significantly clustered axial distribution of the difference angles ($p = 0.043$) with the confidence intervals of the mean angle covering 0°/180° (Fig. R1, left). In fact, in this case we observed that the fish follow the

axis of the magnetic field rather than the pole, resulting in the axial 0°/180° distribution (as opposed to 90°/270° as seen in our other experiments with the 90° rotation, revised Fig. 2). Therefore, while the experiment does not support the presence of random reorientation between two consecutive trials, the significant clustering is consistent with the 0° rotation of the magnetic field that we applied.

In addition, the distribution of the angular differences between the two trials is significantly different from the condition with the 90° rotation (Watson U²: $p < 0.02$, Fig. R1). Please note that here both the differences were computed as 2nd trial – 1st trial to make them directly comparable.

Together with the results obtained for the 90° rotation of the field, this new experiment provides additional evidence that the directional response of the fish observed in our assay depends on the change of the magnetic field direction (either 0° or 90°). The result of the 0° control experiment is now added to the main text (pg. 3) and shown in the revised Fig. S3 of the manuscript.

Fig. R1: Reorientation of zebrafish tested under IR illumination after acclimation to darkness (D-IR) with 0° or 90° MF deflection. For the “0° rotation” condition (left), fish were tested in two groups (tested twice in NE or twice in NW) resulting in a significantly clustered distribution of the angular differences (2nd trial-1st trial) consistent with 0/180°. In comparison, the data from the D-IR group of fish subjected to a 90° deflection (right) shows a distribution of the angular differences (2nd trial-1st trial) that is significantly different from that of the fish tested without rotation of the magnetic field (Watson U²: $p < 0.02$).

The data shown and the tests performed are all done on the change in angle between the rotated conditions. From this I am assuming that the actual orientation of the fish is not significantly biased by the magnetic field at a population level (if this is not the case, then this should be shown). Therefore, the claim seems to be that each individual fish has a preference for its individual orientation within the magnetic field, but this is random with respect to its conspecifics. Of course it is possible that such a system could exist, but what utility could it serve in an evolutionary or navigational context? This should be discussed.

Reply: We thank the reviewer for this interesting point and for giving us the opportunity to discuss it in the revised text. As referenced in the submitted manuscript, directional preference was already shown in groups of zebrafish of the EKK strain when tested under white light (Takebe et al. 2012). However, since the preference was shown to be different for genetic cohorts kept in separate tanks, and because we used a different strain of zebrafish (AB), with different behavioral features as compared to EKK (Lange et al. 2013), we have chosen an intra-subjective design to minimize the sources of variability.

Nevertheless, to analyze whether the orientation to magnetic fields was significantly clustered within the cohort, we now plotted the distribution of the preferences in direction from the first trials under white light illumination (NW and NE normalized to geomagnetic North and combined); making the analysis comparable to the single trials shown by Takebe *et al.* The new results are now shown in the revised Fig. S7 and discussed on pg. 4.

Zebrafish of the D-IR group (tested under IR illumination after acclimation to darkness) showed a significant axial distribution (revised Fig. S7). However, the WL condition showed a trend towards an axial distribution of the 'group preference', that was not significantly non-uniform (revised Fig. S7); while the distribution of angular differences showed that the individual change in preference was significantly clustered, with a mean consistent with the 90° deflection of the applied magnetic field (revised Fig. 2d). The distribution of the other conditions similarly showed trends that did not reach significance. These analyses indicate that the angular difference was more sensitive in detecting a directional preference with respect to the magnetic field direction than the group preferences.

With respect to the larger spread that we observed in the directional preferences at the cohort level (as compared to the distribution of angular differences obtained from retesting individual fish), there are some key points that we would like to consider.

- 1) It is possible that a larger sample size may have revealed a clearer group preference depending on the MF direction, in addition to the change of the preference of the individual fish that we have reported.
- 2) However, the larger dispersion of the cohort preference might be explained by the fact that we tested single fish in isolation and in the absence of spatial cues (other than the magnetic field direction). Under these experimental settings, it was thus not possible for the fish to reference and calibrate the individual preference to other sensory inputs - including the localization of other conspecifics (*e.g.* to enable aligned preference angles to support schooling) - as would likely occur in nature (Herbert-Read 2016).
- 3) Furthermore, evidence is emerging that the spatio-temporal structure of fresh-water schools in nature is not as homogenous as previously believed, but rather characterized by individual spatial preferences within the group where few individuals influence the direction of the escape response of the whole group in a leadership-followership structure (Partridge 1981). Different states can even be exhibited by a group at different time points, varying from a high spatial anisotropy between individuals to a clear polarized structure, which is very much depending on their 'interaction rules' with conspecifics (Partridge 1981).

We identified a preferred direction with respect to the magnetic field direction in fish tested in isolation. Nonetheless, behavioral assays in which cohorts of fish are tested in groups and/or provided with additional spatial cues would be required to draw conclusions on the ecological relevance of the observed magnetoreception for navigation purposes.

As requested by the reviewer, we now discuss these new results in the main text (pg. 4).

2) pERK analysis

The authors compare the number pERK stained neurons in larval medaka either stimulated with an oscillating magnetic field or a stationary one. They then search throughout the brain, reporting on statistical tests for 30 different analyses. Different quantifications are performed in different areas, either counting observable cells, the mean value of the area normalized to tERK, etc. It is a bit concerning that the analysis is not consistent and therefore not equivalent. This is explained in the text as some analyses were not possible in some areas, but why can't mean of the area be used in all regions?

They determine that two analyses of the lateral hindbrain (ratio left/right, control-OMF) and (left-left, control-OMF) show 'significant' differences based on t-tests, with the relatively modest p values of $p = 0.043$ and $p = 0.048$ respectively. This would appear to indicate that the left lateral hindbrain is specifically activated by the oscillating magnetic field. This difference is not obvious in the chosen example images (4b), indicating that if there is an effect, it is fairly subtle. Since the authors perform multiple analyses throughout different areas of the same brains, I believe this suffers from the "multiple comparisons problem" and therefore any conclusion drawn from the $p < 0.05$ result is not statistically sound. If one test is performed, then an alpha value of 0.05 is thought of as acceptable for null hypothesis rejection. However, since 30 tests are performed, the family wise alpha rate is something closer to $(\alpha_T = 1 - (1 - \alpha)^n \text{Tests}) = 0.78$. Therefore, it is quite likely to see one or two false positive results using a 0.05 alpha rate on any individual test, even if no real difference exists in the two populations. Proper multiple comparisons controls, such as the bonferoni correction would set the threshold alpha value at 0.0017, well below the observed values. If the authors want to claim there is a real effect of lateralized activation in the hindbrain, I believe these experiments must be duplicated in a second biological replicate targeted specifically to the lateral hindbrain.

Reply: We thank the reviewer for this feedback on the brain mapping experiment. We have followed all the suggestions and we have clarified our experimental design, hypotheses and results in the revised manuscript (pg. 6).

Following the reviewer recommendations, we have now **(1)** used the mean fluorescence intensity as a metric for all brain regions that were **(2)** selected because they have previously been reported to be involved in magnetoreception, **(3)** corrected for multiple testing and **(4)** have replicated the increased pERK/tERK signal in the lateral hindbrain in an independent cohort of animals.

Specifically for each point:

1) We have now employed the normalized mean fluorescence (pERK/tERK) as a metric for brain activation consistently for all the regions analyzed. In comparison to the cell segmentation approach, such analysis is possible also for brain regions with little cellular resolution.

While the pERK/tERK analysis failed to provide support for a lateralized effect in the hindbrain, it confirmed the overall increase in lateral hindbrain activity observed in fish stimulated with oscillating magnetic field which is now presented in the revised version of Fig. 4.

2) Next, we have now clarified our original hypotheses, focused on investigating only candidate brain areas. First, we apologize for having generated confusion on the selection of these specific brain areas, which we now explain better in the main text (pg. 6). We intended to test only candidate regions with previous known involvement in magnetoreception: 1. Olfactory epithelium, 2. Habenula (left and right should be considered separately because in teleosts these are very different structures based on development, connectivity and function (Bianco and Wilson 2009; Beretta et al. 2012), 3. Pineal gland, 4. Lateral cerebellum and 5. Lateral hindbrain. We originally found a trend of increased activity in the lateral hindbrain (Fig. S8a of the submitted manuscript), and we next asked whether any difference could be observed when the left and the right portions were analyzed separately (submitted manuscript, pg.9). For uniformity, we then decided to assess left and right of each brain region.

Please note that, besides the left and right habenula (which in fish are two separated structures as discussed above) no difference was found between the left and right portion of each brain region in each condition (paired t-test, 2 tails. Oe sham: $p = 0.56$ and oe soMF: $p = 0.79$, lcb sham: $p = 0.17$ and lcb soMF: $p = 0.24$, lhind sham: $p = 0.19$ and lhind soMF: $p = 0.28$). Thus we now pooled the bilateral measurements for these regions. This change is reflected by the revised Fig.4, and in that left-right analysis is no longer reported as Supplementary Figure.

3) As requested by the reviewer, we now also correct for multiple comparisons. We applied the false discovery rate (FDR) method (Benjamini and Hochberg 1995) to control for the proportion of false positives among the rejected tests, when assessing the effect of the soMF on the candidate brain regions. We chose this method because of its increased power as compared to Bonferroni correction since it is controlling for false discovery among the conditions that reached the 0.05 significance level. Thus, this method is less likely to yield false negative result when comparing several conditions, which is the general concern when applying Bonferroni correction. Our analysis with FDR correction yielded a significant increase of the pERK/tERK signal in the lateral hindbrain at the 0.05 significance level. This is now shown in the revised Fig. 4e.

4) In addition, we followed the reviewer's suggestion and replicated the experiment; confirming the significant increase of the mean intensity in the lateral hindbrain in an independent cohort (unpaired t-test, 2 tails, $p = 0.046$, $n = 11$ for sham and $n = 10$ for soMF). No significant difference in the mean intensity between left and right portion of the brain was observed, lhind sham: sham: $p = 0.16$ and lhind soMF: $p = 0.08$). The new dataset is now shown in revised Fig. S9.

3) *Oscillating magnetic field experiments on larval medaka.*

The authors explain that the dual-wrapped design of their coils ensures that a constant amount of current is applied at all times, and thus the same amount of electrical noise. However, it is really the case that there is no audible difference between the oscillating magnetic field condition and the constant condition? One might imagine a change in frequency, which could have an influence of the behaviour of the fish. Can the authors record the resultant noise/vibrations at the dish, and compare between the two conditions? If these differ, then a control addressing the possibility that the fish can hear the oscillation patterns in the coils needs to be addressed.

Reply: We used double-wrapped coils that either work in sham-mode or in field-mode (slowly oscillating magnetic fields, soMF), such that the currents delivered through the power supply is equal in both conditions, as previously described by Kirschvink (1992). We have now expanded upon this in the main text (pg. 5).

In the sham-oscillatory field condition (control) the currents were also oscillatory (and not constant) at the same frequency as in the experimental condition, but just ran in an antiparallel sense. Furthermore, each layer of wires was mechanically fixed in resin to prevent Lorentz-forces from generating movements of the wires which could generate sound or noise. Thus, it is highly unlikely that there were audible differences between the conditions. We have also improved the figure legend of the revised Fig. 3 and the Methods section to explain this more clearly (pg. 10).

In addition, and following the recommendation of the reviewer, we recorded acoustic signals at the center of the arena during soMF or sham mode, using a microphone (Foxnovo, Portable USB 2.0-Kondensator-Mikrofon). As predicted, the measurement did not reveal any clear difference in sound amplitude between the two conditions ($p = 0.54$). Please find below the relative plot and statistical assessment.

Fig. R2: Audio recording at the center of the double wrapped coils running in sham mode (blue) and soMF (red, overlaid) at oscillation of 1Hz. The plot combines two recordings of the two conditions. No significant difference was present between the two conditions (unpaired t-test: $p = 0.54$).

Reviewer #2 (Remarks to the Author):

This is an interesting study testing magnetic orientation in two model species of fish in behavioural experiments and examining brain activity in response to magnetic fields. The study is well conceived, though I disagree with the title and conclusions drawn from the results, i.e. the statement that magnetoreception is light independent.

The data show that the fish respond to a shift of the magnetic field under IR and your (undefined) white light condition (see comments about spectral properties of light below). I agree that these results show that the fish use a light-independent mechanism to orient under IR, and that they likely use the same mechanism under the WL condition. However, this does not mean that they do not also have a light-dependent magnetic compass, as has been shown in newts and birds, which use both systems under different ambient conditions and in different behavioural contexts (see work by Wiltschko, Phillips, Muheim). Birds, for example, if tested under high-intensity light of different spectra, can become disoriented or revert to light-independent magnetic alignment responses. Also, depending on the behavioural context, animals might not use a light-dependent compass, but show an alignment behavior because they try to read magnetic map information. This could be the case also here, so all you can say is that under the experimental conditions that you used and in the behavioural context you tested the fish they used a light-independent magnetic sense. Given the broad phylogenetic occurrence of radical-pair-based magnetoreception in the animal (and plant) kingdom, and that fish have six cryptochrome genes, I would be very careful in suggesting that they do not also have a light-dependent magnetic compass, in addition to light-independent magnetoreception that you describe here.

Reply: We thank the reviewer for recognizing the value of our study. We have now changed the title as well as the main text to clarify that the evidence for a light-independent mechanism in our cohorts of fish does not exclude the presence of also a light-dependent mechanism. The latter may be used in addition or instead of a light-independent mechanism if short-wavelength light is available. We now discuss this and the potential presence of a hybrid system in the revised text and also reflect this point in the revised Fig.1.

We now chose this title for our study: 'Zebrafish and medaka offer insights into the neurobehavioral correlates of vertebrate magnetoreception'

Specific comments:

Analysis of bearing and spatial preference:

I don't understand why the data is binned into 8 sectors, when to start with, you determined both bearing and spatial preference to a precision of one degree. By binning your data, you lose potentially important information, thus I see no advantage of doing so. Also, why do you bin your data in 45 deg sections starting at 0 deg, instead of 0 deg ± 12.5 deg? Figs. 2d-f are very difficult to interpret, since 90 deg is not the center of a bin, which I think is very odd, given that you test your data against the expected 90 deg change. Thus, I recommend to (1) not bin the data at all, or (2) center the data so that 90 deg is the center of one bin (as you appear to have done with spatial preference in Figs. 2 g+h).

Reply: The bearing (BE) is, as correctly stated by the reviewer, determined to a precision of one degree. The data were binned simply for representation, while angular difference and statistical tests were performed on the raw data in order to not lose any information. We understand and apologize for the confusion which might have arisen from the submitted Fig. 2 and Fig. S2. As suggested by the reviewer we now do not bin the bearing data for plotting.

The spatial preference (SP) parameter, on the other hand, is given by the segment in which the fish spent the most of its time (see also Methods section pg. 8). As a consequence, the data as well as the change in preference are naturally binned. We have now reduced the size of the segments from 45° to 30° in order to better resolve the change in preferred position. This new analysis is now also reflected in the revised Fig. S2c',f. As also suggested by the reviewer, we changed the binning of the data for the plot such that 90° is the center of a bin.

Use of V-test: I don't think that you can use the V-test in this context, since your "bearings" are differences between two tests. The authors should carefully check whether they can use a V-test. The alternative would be to use the confidence interval test, as described by Batschelet.

Reply: The V-test and the Rayleigh test in their basic form are applied to a sample consisting of angles (e.g., the bearing of an animal). In most studies using Emlen funnels, however, the bearing of each single bird does not represent just a single angle, but already a first-order mean angle, averaged over at least three trials, so that the group means are grand means from the individual means. For the second-order statistics, it is general practice to calculate an unweighted grand mean for the group, where each individual mean direction is treated as if it had a vector length of unity.

In our case, for each individual fish, we consider the bearing in one condition as reference bearing which we then subtract from the bearing in the other condition. From these first-order differences, we compute the unweighted second-order mean (grand mean). Thus, except for the sign at the first-order level (differences instead of means), our procedure is mathematically equivalent to what is routinely applied in bird orientation studies. Note that, if x and y are two normally distributed random variables, then the two individual variances $\text{var}(x)$ and $\text{var}(y)$ always add up to the total variance, no matter whether $\text{var}(x+y)$ or $\text{var}(x-y)$ is considered.

In detail, for each individual fish, $i \in [1, n]$, we consider the bearing in one condition (NW) as reference bearing which we then subtract from the bearing in the other condition (NE), i.e.,

$$\phi_i = \varphi_i(\text{NE}) - \varphi_i(\text{NW}) \quad (1)$$

From these n (first-order) differences, we determine second-order variables at the group level, which are the group mean angle

$$\bar{\phi} = \tan^{-1}(W/V) \quad (2)$$

and total vector length as

$$n r = R = (V^2 + W^2)^{1/2} \quad (3)$$

where

$$V = (\sum_{i=1}^n \cos \phi_i), W = (\sum_{i=1}^n \sin \phi_i) \quad (4)$$

so that the V' variable for the V-test is obtained as:

$$V' = R \cos(\bar{\phi} - \phi_0) \quad (5)$$

where $\phi_0 = \text{NE} - \text{NW} = 45^\circ - (-45^\circ) = 90^\circ$ (in geographical coordinates).

Note that Eq. (5) is equivalent to

$$V' = \sum_{i=1}^n \cos(\phi_i - \phi_0) \quad (6)$$

which directly shows that ϕ_i values closer to the target ϕ_0 have a larger weight than distant ϕ_i values have. The V-test with its built-in weight function thus circumvents the problem of including the vector length $|d\mathbf{r}|$ of the vector difference between NE and NW bearings,

$$d\mathbf{r}_i = \begin{pmatrix} \Delta x \\ \Delta y \end{pmatrix}_i = \begin{pmatrix} \cos[\varphi_i(\text{NE})] - \cos[\varphi_i(\text{NW})] \\ \sin[\varphi_i(\text{NE})] - \sin[\varphi_i(\text{NW})] \end{pmatrix}$$

Note that $|d\mathbf{r}| \approx 0$ for $\varphi_i(\text{NE}) \approx \varphi_i(\text{NW})$, while $|d\mathbf{r}| = 2$ for $\varphi_i(\text{NE}) \approx 180^\circ + \varphi_i(\text{NW})$, hence inclusion of the vector length of the difference vector as weight would produce a strong bias towards 180° . Therefore, as long as we work with difference angles, not vector differences, we can apply the V-test. Note that our difference angles ϕ_i still represent angular variables because of the 360° periodicity in ϕ_i (e.g, a shift by -180° is the same as one by $+180^\circ$).

In contrast to previous studies on zebrafish (e.g. Takebe et al. 2012), our paired-test design, where individuals were tested twice, allowed us to have a clear hypothesis for the expected mean direction (90° due to 90° declination change of applied field). Because of this *a priori* information about the expected mean, the V-test can be applied, and is also the most powerful especially in the case of scattered data (see e.g. Durand and Greenwood 1958; Batschelet 1981). Thus, this is what we had intended to do when planning the experiments, because we had the 90° deviation defined *a priori* as target by using NW and NE.

Of course, in the case the V-test fails to reject the null hypothesis, one does not know if the data are uniformly distributed or if its mean is much different to the expected value. However, this shortcoming does not affect the case where the V-test does reject the null hypothesis (and which is always the case for the examples shown in Fig. 2 and even in Fig. S4 for the IR experiments without previous acclimation in darkness).

Nonetheless, in addition to the V-test we now report the Rayleigh test for clustering of the data in combination with the 95% confidence interval as suggested by the reviewer. For the conditions under WL and IR with previous acclimation to darkness, we can report significantly clustered distributions according to the Rayleigh test and confidence intervals consistent with a 90° rotation. For the experiments under IR illumination without previous acclimation in darkness, we still obtain significant results with the V-test, but no longer with the Rayleigh test, which demonstrates that the V-test with its additional information (the specified direction) has more power than they Rayleigh test. The lack of significant clustering of the distributions might be due to the sudden change in the illumination condition. This is also the case for the difference in bearing of medaka assessed at 8.5 cm radius, which is thus no longer included in the revised manuscript.

The difference between the NW and NE condition was calculated by subtracting BE(NE) from BE(NW) = BE(NE) – BE(NW), but you tested about half of the animals first under NW and the other first under NE. If true, your differences are incorrect, since the difference in orientation in fish first tested under NE and then NW should be calculated BE(NW) – BE(NE). This might explain the axiality in your data, so please check!

Reply: As correctly stated by the reviewer, we consistently computed the difference as BE(NE)-BE(NW) for all the fish, although half of them were tested first in NE and the other half first in NW. The order of testing was randomized such to minimize effects due to sequence. However, since it was our aim to specifically compare the difference to the expectation of a positive 90° rotation, we computed BE(NE)-BE(NW) for all. This specific calculation allowed us to assess the nature of the response (axial or polar), which might have implications for the type of magnetoreceptive sense and mechanism at play.

In contrast, if the order of testing is taken into account, we would have a mix of BE(NE)-BE(NW) and BE(NW)-BE(NE), and thus only a bimodal (axial) distribution can be expected (due to ± 90°), thus reducing the precision of the prediction. However, as suggested by the reviewer, we now also computed the angular difference according to 2nd trial – 1st trial and observed the same pattern of effects across the conditions as for computation of the angular difference as NE-NW (*i.e.* significant clustering in all cases except in the case of testing under IR illumination without acclimation to darkness, for both species, revised Fig. 2 and Fig. S3). Depending on the opinion of the reviewer and editor we are open to the possibility of including these results in the revised manuscript.

Fig. R3: Angular differences for all experimental conditions computed as 2nd trial-1st trial. We observe a significant clustering of preference changes in the zebrafish (AB) under white light and both zebrafish and medaka (CAB) under IR illumination after acclimation to darkness (upper panel, corresponding to revised Fig. 2d-f), however, not for fish tested under IR but without acclimation to darkness (lower panel, corresponding to revised Fig. S4)

As to the choice of the term ‘reorientation’, we wanted to emphasize that we are re-testing the same individual in an intrasubjective design. In order not to imply a temporal sequence in the term ‘reorientation assay’, we now refer to the assay as a ‘directional preference assay’.

Use of term “GMF”: It is misleading to use the abbreviation for geomagnetic field for artificially deflected fields. Even though they have the same properties as the geomagnetic field, they are artificially created magnetic fields. It is important to distinguish whether, for example, the natural GMF was used as control and an artificially changed field as the experimental field, or whether both control and experimental fields were created with coils. If you use the term GMF you suggest that you used the Earth’s magnetic field, which is not true, so please use another term. Please also give the properties (total intensity, inclination) of your local geomagnetic field.

Use of term “oMF”: Also this term can be misunderstood since some research groups (e.g., Mouritsen lab) use OMF as abbreviation for oscillating RF-fields, which is a different type of oscillating field in regard of its effect on a magnetic sense. If I understand you correctly, your field is a static magnetic field oscillating in intensity and direction at 1 Hz, while an RF-field is a much weaker magnetic field (tens of nT)

oscillating in the MHz frequency along in direction. Thus, your field is a means to test for general sensitivity of animals to magnetic fields, while RF-fields specifically test the radical-pair mechanism. So, even though the hyperactivity in your fish and the one in cockroaches are due to the oscillating fields, the underlying reasons are very different because of the different properties of the two oscillating fields. So, it is very important to make this distinction!

Reply: We thank the reviewer for this point on refining our terminology and now simply use ‘MF’ (with NW and NE for naming the two conditions) for the static magnetic field used in the experiments with mature fish. We choose ‘soMF’ (for slowly oscillating field) for the behavioral test in juveniles. As the reviewer correctly states, our assay is aiming to assess general magnetosensitivity in the fish, in contrast to the RF-field assays performed by others to specifically test the radical-pair mechanism. We now clearly explain our stimulation paradigm used in the locomotor assay (pg. 5).

The horizontal component of the local geomagnetic field was 23.3 μT while the vertical component was 45.0 μT , resulting in a total intensity of 50.6 μT and an inclination of 62.6 deg. We have added these specifications to the Methods section (pg. 8 and revised Fig. S1).

Specification of white light (WL): please provide spectral information, incl. irradiance of your “white light”. The current description does not provide any useful information. How is the light produced (LED, iridescent, halogen), does it include UV, how does white balance look like (short- vs. long-wavelength light)? Work from birds and newts have shown that spectral composition and light intensity are important properties in magnetic compass orientation, thus depending on the properties of the white light used, radical-pair-based magnetoreception might not have been possible for the fish to use, and could be the reason that you did not find light-dependent magnetoreception (see work by Wiltschko, Phillips, Muheim).

Reply: The white light used in our experiment was produced by LEDs (Copic, LED Drawing light table, 22075 663), with an absolute temperature of 5500 K, consistent with full spectrum light. We also measured the spectrum of our light source (Fig R4) and although not including UV, wavelengths are compatible with light-dependent magnetoreception (400-500 nm, Hore and Mouritsen 2016). The maximum light intensity is 4500 lx, while the maximum power is 16.8 W, however, we used ca. 30% of the power in our experiment. Information on the light source and wavelength spectrum is added to the Methods (pg. 10).

Fig. R4: Spectrum of the white light source used in the locomotor activity assay.

Reviewer #3 (Remarks to the Author):

John Phillips

Review for Nature Communications: Myklatun et al. 'Magnetoreception in teleosts is light-independent and influences their locomotor and neural activity.'

The authors report important advances in the development of a new assay to explore the molecular/genetic underpinning of the magnetic sense in teleost fish. There are a number of interesting findings, and considerable potential for larval fish systems to contribute to a better understanding of this novel sensory systems(s).

However, there are several issues with the summary of earlier literature, and more importantly with aspects of the statistical analyzes, that need to be addressed.

Background: With respect to the literature on magnetoreception, several issues need to be clarified. First, the papers cited in Figure 1 should be included in the literature cited. These papers, as well as papers cited elsewhere in the manuscript, are a rather odd selection from the literature. There are no papers cited from the large literature on light-dependent magnetic compass orientation in birds and amphibians, with the exception of the paper by Stapput et al. (2006) that reports magnetic alignment responses in the dark by a migratory bird. What is missing, both in the papers that the authors' cite and in their discussion of the literature, is the large body of evidence indicating that both amphibians and birds have two distinct magnetoreception mechanisms (Phillips 1986, Munro et al. 1997) --- a light-dependent, axial/inclination magnetic compass (Phillips & Borland 1992, Wiltschko et al. 1993, Deutschlander et al. 1999) and a non-light-dependent, polarity sensitive 'map' detector involved in sensing subtle spatial variation in the magnetic field used to derive a map or geographic position sense (Phillips & Borland 1994, Munro et al. 1997, Phillips et al. 2002).

By only citing the paper by Stapput et al. from the long list of behavioral experiments carried out in the Wiltschko lab (the other citations for birds in Fig 1 are for neurophysiological experiments), the authors give the impression that birds have only a non-light-dependent magnetoreception mechanism, which is clearly not the case (Wiltschko et al. 2010).

Likewise, the authors' citations of behavioral evidence in rodents in Figure 1 are primarily from molerats tested in the dark, and a somewhat offbeat paper by Prato et al. The finding that molerats can orient in the dark, and appear to have a non-light-dependent magnetoreception mechanism, is not surprising given that these rodents are adapted to live in aphotic subterranean habitats. There is no mention of neurophysiological/biochemical evidence for a light-dependent magnetoreception mechanism in the retinas of murine rodents, or of behavioral evidence for a well-developed magnetic compass that is sensitive to low-level radio frequency fields (Deutschlander et al. 2003, Olcese 1990, Muheim et al. 2006, Phillips et al. 2013, Malkemper et al. 2015).

Given the presence of both magnetite-based magnetoreceptors in molerats and light-dependent, RF-sensitive magnetoreceptors in murine rodents, it would not be surprising if murine rodents, and perhaps other mammals, also have dual mechanisms, and that this might also be the case in fish. To date there has been no attempt to investigate whether murine rodents or teleosts have a magnetite-based magnetoreception mechanism (e.g.) under conditions where input from an RF-sensitive, light-dependent magnetic compass would be eliminated or downgraded. A more thorough discussion of the literature is needed because the findings reported by the authors point to the presence of a non-light-dependent magnetoreception mechanism in teleosts, but provide no evidence about whether a second, light-

dependent mechanism is also present. This is important because the neural activity studies of larval medaka were carried out under white light. Therefore, although the authors present evidence for the involvement of a non-light-dependent magnetoereception mechanism in mediating some of the behavioral responses of adult fish and larval medaka, their findings do not distinguish between the possibilities that the increased neural activity in larval medaka is due to a light-dependent mechanism, a non-light-dependent mechanism, or both.

Reply: First, we apologize for not having included all references from Fig.1 in the literature list; we have now added all of them in the revised manuscript. As requested by the reviewer, we also revised Fig.1 and added references to present the literature in a more balanced way, in order to also include evidence for the presence of dual mechanism in birds and amphibians. Please note that the revised Fig. 1 no longer contains references to works referring to only the neuronal correlates of magnetoreception.

Regarding the literature on rodents, as correctly pointed out by the reviewer, wood mice seem to be sensitive to oscillating magnetic fields (Malkemper et al. 2015), indicative of a radical pair mechanism. However, it is interesting to note that studies exist showing that the magnetic field-dependent orientation behavior of mole rats, which not surprisingly occurs in darkness given their evolutionary adaptation to subterranean environments, is unperturbed by oscillating high-frequency fields (Thalau et al. 2006). It thus seems unclear whether rodents have two different systems depending on species or if they also have a dual mechanism.

As requested, this evidence is now reflected by the literature cited in the revised Fig.1 as well as in the introduction and discussion. Specifically, we make explicit that, although the adult fish seem to use a light-independent mechanism in our assay, this does not exclude the possible presence of also a light-dependent mechanism that can be used if short-wavelength light is available. However, investigating this possibility was beyond the scope of the paper.

We have also changed the title to: “Zebrafish and medaka offer insights into the neurobehavioral correlates of vertebrate magnetoreception”.

Should the authors choose to carry out future research to distinguish between these alternative possibilities (which is not necessary for the current manuscript), they should keep in mind that electromagnetic shielding should be used to screen out low-level radio frequency interference (e.g., radiated by video cameras, stimulus coils, microscope lights, as well as computers and other equipment located nearby) that has been shown to increase the variability of responses mediated by putative RPMs (Muheim et al. 2006, Vacha et al.2009, Phillips et al. 2013, Engles et al. 2014, Landler et al. 2015, Malkemper et al. 2015).

Reply: We thank the reviewer for this constructive comment on how to reduce this source of variability in future studies.

There are several issues with the data and statistical analysis that should be addressed. The ‘V-test’ is no longer widely used in the spatial literature. This is because the test statistic (the component of the mean vector that is in the expected direction) is dependent both on the mean vector bearing and mean vector length—i.e., for a given sample size, the same level of significance can be obtained from a scattered distribution with a (relatively weak) mean vector that coincides with the expected direction and a tightly clustered distribution with a very strong mean vector with a large deviation (up to 70-80 degrees) from the expected direction, i.e., both could have the same component in the expected direction. There was a lengthy discussion of this in the literature some years ago, and the consensus was that a Rayleigh test should be used to determine if there is greater clustering in the distribution of bearings than would be expected by chance. If so, a 95% confidence interval for the mean vector bearing can then be used to determine if the mean vector bearing is consistent with the expected direction. In the case of a tightly

clustered distribution of bearings with a mean vector that deviates substantially (e.g., by 40-50 degrees or more, but less than 90 degrees) from the expected direction, a possible conclusion is that there is strong clustering in the distribution of bearings, but that this orientation deviates significantly from the expected direction.

Reply: We thank the reviewer for this point and have now included both the Rayleigh test and confidence intervals (as also suggested by reviewer 2). Generally speaking, as the reviewer correctly states, it is true that the V-test has not been used much recently in the avian magnetic orientation community, which we think is due also to a shift in focus from “demonstration of magnetic orientation” to “exploration of underlying mechanisms” using lesions, radiofrequency magnetic fields, and spectral light conditions, which specifically test for disorientation effects, but not for (re)orientation into a specific direction.

We agree on the limited usefulness of the V-test when applied to strongly concentrated bearings. This is however not the case with our more scattered data. The Rayleigh test is said to be “not uniformly most powerful” (Durand and Greenwood 1958) because it does not test against a specific alternative. When there is an *a priori* hypothesis about a preferred direction or axis, the V-test is the test of choice (unless the data are strongly concentrated). For example, Gagliardo, Odetti, and Ioalè (2001) correctly used the V-test to test if bearings of pigeons in an arena were significantly oriented toward a meaningful direction that was known in advance (*i.e.*, the home direction).

Moreover, the V-test is a single test, whereas a criterion based on Rayleigh test plus the confidence intervals represents a case of two tests applied to the same sample, which from the point of view of statistical test theory is not ideal. Using the confidence interval criterion alone, we observe that the 95% range includes the 90° (and similarly the 0° target for the additional experiment requested by reviewer 1), see revised Fig. 2. As already mentioned in the response to reviewer 2 (page 9 of this document), for the experiments under IR illumination without previous acclimation in darkness, we still obtain significant results with the V-test, but no longer with the Rayleigh test, which demonstrates that the V-test with its additional information (the specified direction) has more power than the Rayleigh test. We attribute the lack of significant clustering of the distribution to the sudden change in the illumination condition representing a strong stimulus that is independent of the magnetic field (please see our revised Fig. 2 and Fig. S4). Please also note that with the new criteria our analysis of bearing at the larger radius (8.5 cm, which was originally applied to medaka) did no longer yield a significant result (apart from the V-test), which is why we decided to not include those data in the revised manuscript.

With respect to the distributions showing axially bimodal responses. By plotting each data point twice ('double plotting'), a practice that has been used occasionally in the literature, the authors indicate that the direction of the each response along the 'axis of response' is unimportant. However, this needs to be clearly stated in the methods and/or figure caption; my apologies if it was stated somewhere in the manuscript and I missed it. Each of the distributions should be plotted with only one data point for each bearing (regardless of whether the bearings are unimodally or bimodally distributed) to help avoid confusing readers who might find a disconnect between the sample size and the number of bearings when the bearings for the axially bimodal distributions are 'double plotted' (e.g., Fig 2d,e; S3a,d; S6a).

Reply: We apologize for not having clearly stated that we used the double plotting for the axial distributions. We now plot only the real data points in the figure, as requested, but add the double arrow calculated by doubling the angles and to indicate the axial symmetry. See e.g. the revised Fig. 2.

Notice also that in Fig S6, the number of bearings is greater than the sample size given, although in this case the discrepancy is not due to double-plotting. Also in Fig 2h, the sample size agrees with the

number of bearings, but the distribution of bearings is symmetrical (consistent with double plotting) so might be worth double-checking.

Reply: We thank the reviewer for this observant comment. Indeed there was a mistake in reporting the number of observations in these plots which is now corrected. We also carefully went through all our data again, so please note also the corrected number for AB WL in revised Fig 2.

Once the numbers of bearings vs. samples sizes have been corrected in the axially symmetrical distributions, it is very important that the authors show a figure of the actual distribution of bearings in each figure, i.e., without double plotting. For distributions that are axially symmetrical, a second distribution can be included with the bearings doubled (each 'doubled' bearing plotted only once) and a unimodal mean vector bearing calculated from the distribution of doubled bearings (Note: this is not the same as double-plotted bearings) to show if the bimodal distribution is non-randomly distributed. It's important to emphasize yet again the importance of plotting the distribution of individuals bearings for each experimental treatment (i.e., whether or not the distribution shows axial symmetry), so readers can accurately assess the sample sizes, and visually compare the distributions under different experimental treatments.

Reply: We now represent the data with axial symmetry by only plotting the actual bearing in black, while the axial mean vector is represented by a double arrow, the direction of which is calculated by doubling the angles. This will allow readers to accurately assess the sample size and the axiality of the distributions.

I would also encourage the authors to use a two-sample test to compare the distributions of doubled bearings in the different treatment conditions, e.g., with a two-sample test (e.g., Watson U₂). It is quite possible that the responses of adult zebrafish under WL are significantly different from those under IR and D-IR, which may turn out to be interesting given the evidence that other vertebrates have dual light-dependent and non-light-dependent magnetoreception mechanisms (see above). There may also be a difference between the responses of adult medaka under IR and D-IR. Needless to say, interpretation of the patterns of neural activity depends not only on their being an effect, but also on whether or not there are multiple effects and, perhaps, multiple magnetoreception mechanisms involved.

Reply: We thank the reviewer for this very interesting suggestion. With this additional analysis of the data, we can clearly see the importance of the acclimation period. In the case of both zebrafish and medaka we observed a significant change of their orientation according to the rotation of the magnetic field only when acclimated to darkness prior to testing under IR illumination. Indeed for zebrafish, the distribution of the change in orientation in IR is significantly different from that in the D-IR condition (IR vs D-IR(90), Watson U²: $p < 0.02$). Furthermore, the D-IR condition for zebrafish tested with a 90° rotation of the field is also significantly different from the distribution of those tested in WL (WL vs. D-IR(90), Watson U²: $p < 0.05$).

For medaka, there was no significant difference when comparing the distributions tested under IR without and with acclimation to darkness (IR vs D-IR, Watson U²: $0.1 < p < 0.2$).

One final point of information—the authors mention that some populations of medaka migrate back and forth between marine and freshwater habitats. I assumed that larval zebrafish and medaka were both tested in fresh water, but couldn't find this stated anywhere in the manuscript. If the larval medaka (but obviously not larval zebrafish) were tested in salt or brackish water, an additional control(s) should be carried out to make sure that the larvae were not responding to an electrical field induced by the 1 Hz oscillating magnetic field.

Reply: As correctly assumed by the reviewer, all tests were performed in fresh water, as both species are raised and kept in a fresh water aquarium system. This information is now added to the Methods section (pg. 10).

Please feel free to contact me by e-mail (Phillip@vt.edu) to discuss any or all of these issues.

John Phillips

Literature cited

- Deutschlander, M.E., S.C. Borland & J.B. Phillips 1999. Extraocular magnetic compass in newts. *Nature* 400: 324-325.
- Deutschlander, M.E., M.J. Freake, S.C. Borland, J.B. Phillips, L.E. Anderson & B.W. Wilson 2003. Learned magnetic compass orientation by the Siberian hamster, *Phodopus sungorus*. *Anim. Behav.* 65: 779-786.
- Engels, S. et al. 2014. Anthropogenic electromagnetic noise disrupts magnetic compass orientation in a migratory bird. *Nature*
- Landler, L., Painter, M.S., Youmans, P.W., Hopkins, W.A., Phillips, J.B. 2015. Radio frequency field affects association of magnetic field with novel surroundings in yearling snapping turtles (*Chelydra serpentina*). *PLoS ONE* 10(5): e0124728. doi:10.1371/journal.pone.0124728.
- Malkemper, E. P., S. H. K. Eder, S. Begall, J. B. Phillips, M. Winklhofer, V. Hart & H. Burda 2015. Magnetoreception in the wood mouse (*Apodemus sylvaticus*): influence of weak frequency-modulated radio frequency fields. *Sci. Rep.*, 9917; doi:10.1038/srep09917.
- Muheim, R, N.M. Edgar, K.S. Sloan & J.B. Phillips 2006. Magnetic compass orientation in C57BL6 mice. *Learn. Behav.* 34: 366-373.
- Munro, U., J.A. Munro, J.B. Phillips, R. Wiltschko & W. Wiltschko 1997. Evidence for a magnetite based navigational 'map' in birds. *Naturwissenschaften* 84:26-28.
- Olcese, J. M. The neurobiology of magnetic field detection in rodents. *Prog Neurobiol* 35, 325–330 (1990).
- Phillips, J.B.. 1986. Two magnetoreception pathways in a migratory salamander. *Science* 233:765 767
- Phillips, J.B. and S.C. Borland. 1992. Behavioral evidence for the use of a light dependent magnetoreception mechanism by a vertebrate. *Nature* 359:142 144
- Phillips, J.B. and S.C. Borland. 1992. Magnetic compass orientation is eliminated under near infrared light in the eastern red spotted newt *Notophthalmus viridescens*. *Anim. Behav.* 44:796 797
- Phillips, J.B. and S.C. Borland. 1994. Use of a specialized magnetoreception system for homing. *J. Exp. Biol.* 188:275 291
- Phillips, J.B., K.A. Adler and S.C. Borland. 1995. Navigation by an amphibian. *Anim. Behav.* 50: 855 858.
- Phillips, J.B., M.J. Freake, J.H. Fischer, S.C. Borland 2002. Behavioral titration of a magnetic map coordinate. *J. Comp. Physiol.* 188: 157-160.
- Phillips, J.B, S.C. Borland, M. J. Freake, J. Brassart & J.L. Kirschvink 2002. "Fixed Axis" Magnetic Orientation by an Amphibian: Non Shoreward Oriented Compass Orientation, Misdirected Homing, or Positioning of a Map Detector in a Consistent Alignment Relative to the Magnetic Field? *J. Exp. Biol.* 205: 3903-3914.
- Phillips, J.B., P.W. Youmans, R.Muheim, K. Sloan, L Landler, M.S. Painter, & C. Anderson 2013. Rapid learning of magnetic compass direction by C57BL/6 mice in a "plus" water maze. *PLOS ONE* 8: e73112.
- Ritz, T., P. Thalau, J.B. Phillips, R. Wiltschko & W. Wiltschko 2004. Avian Magnetic Compass: Resonance Effects Indicate a Radical Pair Mechanism. *Nature* 429: 177-180.
- Stapput, K., P. Thalau, R. Wiltschko & Wiltschko, W. 2006. Orientation of birds in total darkness. *Cur. Biol.* 18: 602-606.

Wiltschko, W., Munro, U., Ford, H. & Wiltschko, R. 1993 Red light disrupts magnetic orientation of migratory birds. *Nature* 364, 525–527.

Wiltschko, R., Stapput, K., Thalau, P. & Wiltschko, W. 2010. Directional orientation of birds by the magnetic field under different light conditions. *J. R. Soc. Interface* 7, S163-77

Vacha, M., Puzova, T. & Kviclova, M. Radio frequency magnetic fields disrupt magnetoreception in American cockroach. *J Exp Biol* 212, 3473–3477 (2009).

See also:

Phillips, J. B., P.E. Jorge, & R. Muheim 2010. Light-dependent magnetic compass orientation in amphibians and insects: candidate physiological and molecular mechanisms *J. Roy. Soc. Interface* 7: S241-256.

Phillips, J.B., R. Muheim, & P. E. Jorge 2010. A behavioral perspective on the biophysics of the light-dependent magnetic compass: a link between directional and spatial perception? *J. Exp. Biol.* 213: 3247-3255.

References:

Batschelet, Edward. 1981. *Circular Statistics in Biology*. Academic Press.

Benjamini, Yoav, and Yosef Hochberg. 1995. "Controlling the False Discovery Rate: A Practical and Powerful Approach to Multiple Testing." *Journal of the Royal Statistical Society. Series B, Statistical Methodology* 57 (1). [Royal Statistical Society, Wiley]: 289–300.

Beretta, C. A., N. Dross, J. A. Guiterrez-Triana, S. Ryu, and M. Carl. 2012. "Habenula Circuit Development: Past, Present, and Future." *Frontiers in Neuroscience* 6 (51). <https://www.ncbi.nlm.nih.gov/pmc/articles/PMC3332237/>.

Bianco, Isaac H., and Stephen W. Wilson. 2009. "The Habenular Nuclei: A Conserved Asymmetric Relay Station in the Vertebrate Brain." *Philosophical Transactions of the Royal Society of London. Series B, Biological Sciences* 364 (1519): 1005–20.

Durand, David, and J. Arthur Greenwood. 1958. "Modifications of the Rayleigh Test for Uniformity in Analysis of Two-Dimensional Orientation Data." *The Journal of Geology* 66 (3): 229–38.

Gagliardo, A., F. Odetti, and P. Ioalè. 2001. "Relevance of Visual Cues for Orientation at Familiar Sites by Homing Pigeons: An Experiment in a Circular Arena." *Proceedings. Biological Sciences / The Royal Society* 268 (1480): 2065–70.

Herbert-Read, J. E. 2016. "Understanding How Animal Groups Achieve Coordinated Movement." *Journal of Experimental Biology*. <http://jeb.biologists.org/content/219/19/2971.abstract>.

Hore, P. J. & Mouritsen, H. The Radical-Pair Mechanism of Magnetoreception. *Annu. Rev. Biophys.* **45**, 299–344 (2016).

Kirschvink, J. L. 1992 Uniform magnetic fields and double-wrapped coil systems: improved techniques for the design of bioelectromagnetic experiments. *Bioelectromagnetics* **13**, 401–411.

Lange, Merlin, Frederic Neuzeret, Benoit Fabreges, Cynthia Froc, Sebastien Bedu, Laure Bally-Cuif, and William H. J. Norton. 2013. "Inter-Individual and Inter-Strain Variations in Zebrafish Locomotor Ontogeny." *PloS One* 8 (8): e70172.

Malkemper, E. Pascal, Stephan H. K. Eder, Sabine Begall, John B. Phillips, Michael Winklhofer, Vlastimil

- Hart, and Hynek Burda. 2015. "Magnetoreception in the Wood Mouse (*Apodemus Sylvaticus*): Influence of Weak Frequency-Modulated Radio Frequency Fields." *Scientific Reports* 4 (April): 9917.
- Partridge, B. L. 1981. "Internal Dynamics and the Interrelations of Fish in Schools." *Journal of Comparative Physiology A: Neuroethology*.
<http://www.springerlink.com/index/G525712X41810132.pdf>.
- Stewart, Adam, Jonathan Cachat, Keith Wong, Siddharth Gaikwad, Thomas Gilder, John DiLeo, Katie Chang, Eli Utterback, and Allan V. Kalueff. 2010. "Homebase Behavior of Zebrafish in Novelty-Based Paradigms." *Behavioural Processes* 85 (2): 198–203.
- Takebe, Akira, Toshiaki Furutani, Tatsunori Wada, Masami Koinuma, Yoko Kubo, Keiko Okano, and Toshiyuki Okano. 2012. "Zebrafish Respond to the Geomagnetic Field by Bimodal and Group-Dependent Orientation." *Scientific Reports* 2 (October): 727.
- Thalau, Peter, Thorsten Ritz, Hynek Burda, Regina E. Wegner, and Roswitha Wiltschko. 2006. "The Magnetic Compass Mechanisms of Birds and Rodents Are Based on Different Physical Principles." *Journal of the Royal Society, Interface / the Royal Society* 3 (9): 583–87.

Reviewers' comments:

Reviewer #1 (Remarks to the Author):

The authors have performed the additional experiments I requested, which have all supported their initial conclusions that adult zebrafish re-orient under a re-orienting magnetic field, and that there is activation of the larval medaka lateral hindbrain when exposed to a slowly oscillating magnetic field. Therefore, I have no further concerns that need addressing.

Reviewer #2 (Remarks to the Author):

I appreciate the clarifications and improvements made in the revised manuscript which, together with the new control data, make it much easier to assess the biological significance of the findings.

I am fine with the explanation to use the angular difference $BE(NE) - BE(NW)$, however I do not agree with the conclusion that the responses shown by the zebrafish under the light condition "was consistent with the 90 deg deflection of the MF" (lines 109-110). Only four of the twelve individuals shifted their orientation +90 deg, while the other eight fish shifted by -90 deg (Fig 1d). I still don't think that the V-test should be used, for the reasons highlighted by reviewer 3, but if it is used nevertheless to test for the +90 deg shift (and not an axial shift), it should be applied to the unimodal distribution of bearings. I assume that in the case of Fig. 1d, the V-test was applied on the axial data, thereby testing for significance along the 90 deg axis. From a biological point of view, testing for an axial response is totally fine, as there are plenty of studies showing axial responses to MF changes, but this has to be clearly stated in the text. Interestingly, this axial response in the light condition compared to the unimodal response in the dark condition could indicate the presence of two different magnetic mechanisms, an axially sensitive, light-dependent magnetic compass and a polarity-based alignment response in the dark.

I am satisfied with the other changes to the manuscript and have no further comments.

Reviewers' comments:

Reviewer #1 (Remarks to the Author):

The authors have performed the additional experiments I requested, which have all supported their initial conclusions that adult zebrafish re-orient under a re-orienting magnetic field, and that there is activation of the larval medaka lateral hindbrain when exposed to a slowly oscillating magnetic field. Therefore, I have no further concerns that need addressing.

Reply: We thank the reviewer again for the suggested additional experiments, which greatly helped us to improve the manuscript.

Reviewer #2 (Remarks to the Author):

I appreciate the clarifications and improvements made in the revised manuscript which, together with the new control data, make it much easier to assess the biological significance of the findings.

I am fine with the explanation to use the angular difference $BE(NE) - BE(NW)$, however I do not agree with the conclusion that the responses shown by the zebrafish under the light condition "was consistent with the 90 deg deflection of the MF" (lines 109-110). Only four of the twelve individuals shifted their orientation +90 deg, while the other eight fish shifted by -90 deg (Fig 1d). I still don't think that the V-test should be used, for the reasons highlighted by reviewer 3, but if it is used nevertheless to test for the +90 deg shift (and not an axial shift), it should be applied to the unimodal distribution of bearings. I assume that in the case of Fig. 1d, the V-test was applied on the axial data, thereby testing for significance along the 90 deg axis. From a biological point of view, testing for an axial response is totally fine, as there are plenty of studies showing axial responses to MF changes, but this has to be clearly stated in the text. Interestingly, this axial response in the light condition compared to the unimodal response in the dark condition could indicate the presence of two different magnetic mechanisms, an axially sensitive, light-dependent magnetic compass and a polarity-based alignment response in the dark.

I am satisfied with the other changes to the manuscript and have no further comments.

Reply: The reviewer is correct in assuming that for Fig. 1d, the V-test was applied for the $\pm 90^\circ$ axis and we have now clarified this in the main text and method section:

"We found that under illumination with white light (WL, Fig. 2b) zebrafish significantly changed their bearing such that the distribution of angular differences between the two magnetic conditions (NE-NW) showed a mean axis which was consistent with the 90° deflection of the MF (Fig. 2d)." (page 3)

We thank the reviewer for the interesting point of discussion regarding the axial response of zebrafish observed in light in contrast to the polar distribution in darkness. Following the reviewer's suggestion, we now state that this may indicate that two different mechanisms might be present:

"Interestingly, the distribution of the individual angular differences in the D-IR group seemed to be polar, and was significantly different from that observed in light (WL vs. D-IR, Watson U^2 : $p < 0.05$), indicating that two different mechanisms might be in play." (page 3)

REVIEWERS' COMMENTS:

[None provided to the authors - Reviewer 2 confirmed that s/he was happy with the latest response.]